# Structural basis of GAIN domain autoproteolysis and cleavage-resistance in the adhesion G-protein coupled receptors

Fabian Pohl [1,2,6], Florian Seufert [3,6], Yin Kwan Chung [2], Robin Schick [1], Björn Kieslich [1], Torsten Schöneberg [2], Tobias Langenhan [2,4,5] ✉, Peter W. Hildebrand [3] ✉ & Norbert Sträter [1] ✉

The GAIN domain is a hallmark of adhesion G protein-coupled receptors (aGPCRs), as this extracellular domain contains an integral agonistic sequence (*Stachel*) that activates the receptor by binding to its 7-transmembrane helix (7TM) domain. Many aGPCRs undergo autoproteolytic cleavage at the GPCR proteolysis site (GPS), which has a canonical H-X-S/T sequence motif located within the GPCR autoproteolysis-inducing (GAIN) domain. Here we present the crystal structure of the Hormone Receptor (HormR) and GAIN domains of ADGRB2/BAI2. The protein was not cleaved at the GPS, despite possessing an HLS sequence. Through structural comparisons and molecular dynamics (MD) simulations, we identify determinants that contribute to autoproteolytic activity beyond the H-X-S/T motif. Specifically, we highlight a T-shaped π–π interaction between the histidine base of the H-X-S/T motif and a phenylalanine residue that is highly conserved in cleavage-competent aGPCRs. This interaction is critical for properly positioning the imidazole group of the histidine to deprotonate the alcohol nucleophile. Disruption of this interaction reduces autoproteolytic activity in the ADGRL1 and introduction of the phenylalanine restores cleavage competence in the otherwise non-cleavable ADGRB3 upon expression in HEK293 cells. In addition, the poorly conserved and flexible flap regions flanking the GPS also contribute to full autocleavage activity.

The GPCR autoproteolysis-inducing (GAIN) domain is a key feature of adhesion G-protein coupled receptors (aGPCRs), which distinguishes them from all other GPCRs[1]. The GAIN domain bears a tethered agonist, referred to as the *Stachel*, which, when released from the GAIN domain, can bind to the orthosteric binding site within the 7-transmembrane helix (7TM) domain, common to all GPCRs, leading to receptor activation[2,3]. Crystal structures of the GAIN domains of the human ADGRB3/BAI3 (B3) and rat ADGRL1/LPHN1 (L1) revealed a fold comprising an α-helical subdomain A followed by the C-terminal subdomain B, which consists of a twisted β-sandwich including eleven β-strands with additional short α-helices and β-strands located within loop regions[4]. Additional crystal structures of the murine ADGRG1/GPR56 (G1)[5], zebrafish ADGRG6/GPR126 (G6)[6], human ADGRG3/GPR97 (G3)[7], and ADGRF1/GPR110 (F1)[8] as well as cryo-EM structures of ADGRE5/CD97 (E5)[9] and ADGRL3/Latrophilin3 (L3)[10] confirmed the overall architecture of the GAIN domain fold (with differences in the

[1]Institute of Bioanalytical Chemistry, Leipzig University, Leipzig, Germany. [2]Rudolf-Schönheimer-Institute of Biochemistry, Leipzig University, Leipzig, Germany. [3]Institute of Medical Physics and Biophysics, Leipzig University, Leipzig, Germany. [4]Comprehensive Cancer Center, Central Germany, Leipzig University, Leipzig, Germany. [5]Institute of Biology, Faculty of Life Sciences, Leipzig University, Leipzig, Germany. [6]These authors contributed equally: Fabian Pohl, Florian Seufert. ✉e-mail: tobias.langenhan@gmail.com; peter.hildebrand@medizin.uni-leipzig.de; strater@bbz.uni-leipzig.de

**Fig. 1 | Model of the catalytic mechanism of GPS autoproteolysis with an HLT motif[14].** Starting from the uncleaved triad **1** the histidine acts as a base, abstracting a proton from the threonine (or alternatively from a serine), resulting in an alkoxide nucleophile attacking the carbonyl group of the scissile peptide group. A tetrahedral intermediate with the five-membered oxazolidine ring **2** is formed. Upon protonation the ring nitrogen leaves, completing an N-O acyl shift to the ester intermediate **3**, which undergoes hydrolysis to the cleaved GPS triad **4**. This mechanism describes *cis*-autoproteolysis of GAIN domains with a canonical H-X-S/T motif. B a general base, CTF/NTF C-/N-terminal fragment, GPS GPCR proteolysis site.

length of the helical subdomain A). Proteolytic cleavage of aGPCRs was first indicated for human E5, which is processed into two non-covalently associated fragments[11]. Later, post-translational processing of murine L1[12] and ADGRE4/EMR4[13] was demonstrated. The GAIN domain is sufficient for autoproteolytic cleavage of the peptide bond N-terminally of the S/T residue within the canonical H-X-S/T sequence motif of the GPCR proteolysis site (GPS)[4]. The histidine of this sequence motif acts as a base, which deprotonates the serine or threonine nucleophile to activate it for attack at the peptide bond between the X and S/T residues, thereby initiating proteolytic cleavage[14] (Fig. 1).

The polypeptides generated by autoproteolysis at the GPS are designated as N-terminal and C-terminal fragments (NTF and CTF)[15]. The agonistic *Stachel* sequence follows C-terminally of the cleavage site, with the serine or threonine as the first residue of this sequence. CTF constructs are typically constitutively active, and short synthetic *Stachel* peptides activate CTF constructs that lack the *Stachel*, resulting in an activation model involving release of the *Stachel* peptide from the GAIN domain and binding to the 7TM domain[2,3,16,17]. Notably, the *Stachel* forms a β-strand within the β-sandwich of GAIN subdomain B and appears to be tightly bound by the β-sheet and hydrophobic side-chain interactions, which stabilize the NTF-CTF complex after cleavage[4,11,13] even when mounted on the plasma membrane[18]. The structural basis of the tethered agonist activation model was recently complemented by cryo-EM structures of the CTF of ADGRD1/GPR133[19], F1[19], ADGRG2/GPR64[20], G3[21], ADGRG4/GPR112[20,22], G1[23], ADGRD1/GPR133[24], ADGRG5/GPR114[24], and ADGRL3/LPHN3 (L3)[23,25]. These structures revealed a conserved binding mode of the *Stachel* sequence to the 7TM domain. Cryo-EM structures of the GAIN-7TM constructs have been obtained for E5[9] and L3[10]. Both structures show constrained flexibility in the relative orientations of the two domains, which resulted in low resolution of the GAIN domain (E5) or the 7TM domain (L3).

However, not all aGPCRs undergo autoproteolytic cleavage, and ten of the 32 human aGPCRs lack the H-X-S/T motif. G5 contains an HLT motif at the GPS, is cleavage-incompetent upon expression in COS-7 cells, but exhibits *Stachel*-mediated GPCR signaling[26]. While the GAIN domain structures of L1[4], G1[5], G3[7], and G6[6] were determined in the cleaved state, B3 was obtained in an uncleaved state[4]. However, B3 and the closely related ADGRB1 (B1) receptor contain an RLS motif at the GPS. An arginine side chain should not be able to act as a base and deprotonate the serine nucleophile for autoproteolysis[14] (Fig. 1). Nevertheless, for ADGRB1 and B3, CTF-size fragments were detected in mouse brain lysates via antibodies against the cytosolic region, but not upon expression in HEK293 cells[4]. Based on the close homology of B2, which contains a canonical HLS motif at the GPS, to B3, we became interested in studying the B2 structure and cleavage state to better understand the structural determinants and mechanism of GPS cleavage.

Here we present the crystal structure of the B2 GAIN domain, which we obtained in an uncleaved state. We find that the catalytic H-X-S/T motif is not sufficient for efficient autoproteolysis. MD simulations are used to study the ensemble of conformations accessible to the GPS catalytic residues in comparison to a model of the uncleaved L1 GAIN domain. By comparing the MD structures of these two receptors and models of cleavage-competent and cleavage-resistant receptors, a π–π interaction of the histidine base with a highly conserved phenylalanine side chain is identified, which orients the histidine base to accept a proton from the serine/threonine nucleophile. The relevance of this interaction is confirmed by mutational studies. The GPS is flanked by poorly conserved and flexible regions termed "flaps". Exchange chimera of these flaps demonstrates their importance for full auto-cleavage activity.

## Results

### Crystal structure of the B2 HormR and GAIN domains

The structure of the extracellular region (ECR) of *h*B2, as predicted by AlphaFold2 (AF2)[27], consists in N→C-terminal order of a complement C1r/C1s, Uegf, Bmp1 (CUB) domain, four thrombospondin type 1 (TSP1) domains, a hormone receptor (HormR) domain, and a GAIN domain (Supplementary Fig. 1B). No folded domains are predicted for the intracellular region of the receptor. Stable interdomain interactions are predicted to exist only between the HormR and GAIN domains (Supplementary Fig. 1A). For crystal structure analysis, we expressed and purified a construct consisting of the HormR-GAIN domains (*h*B2-HG) in HEK293S cells (Supplementary Figs. 1C, 2). The protein exhibited favorable monodispersity and a high melting temperature of 76.60 ± 0.08 °C (Supplementary Fig. 3). It could be crystallized

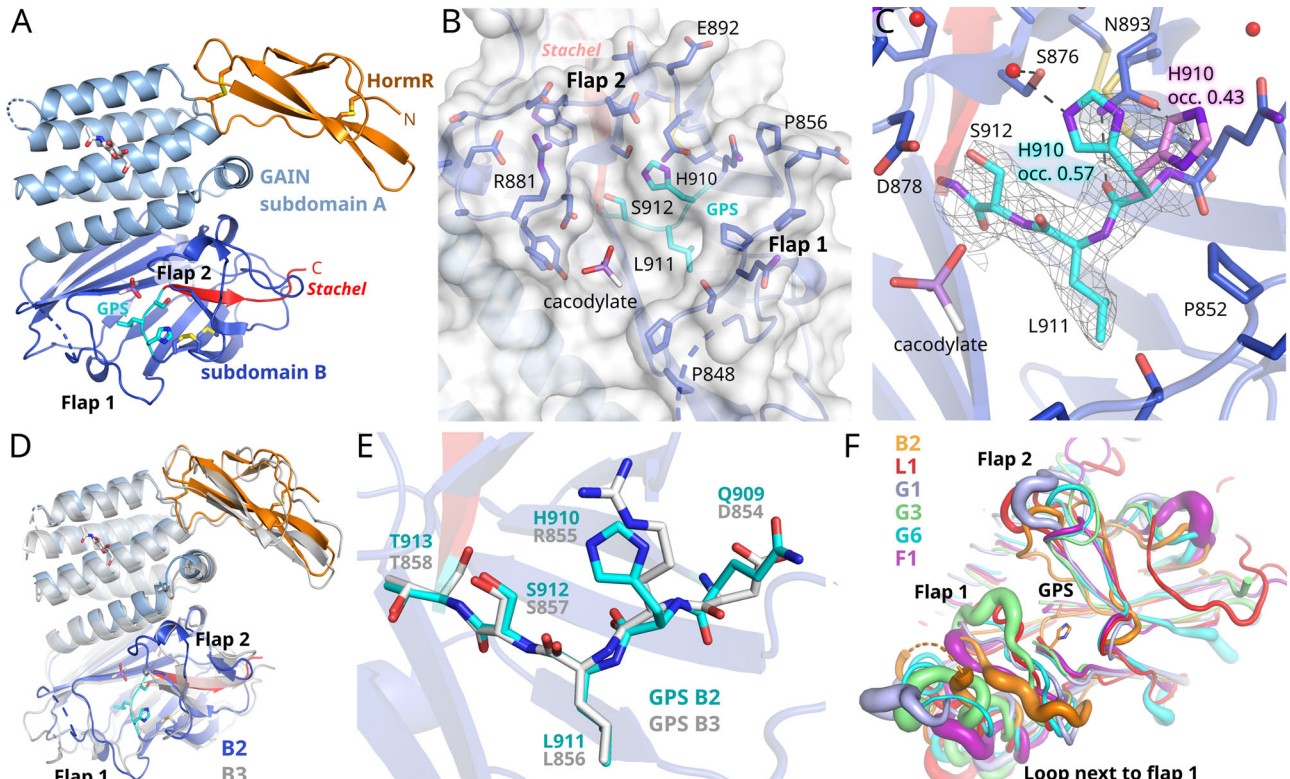

**Fig. 2 | Crystal structure of the B2 HormR and GAIN domains and structural comparisons. A** Overview of the domain structures. Two disulfide bridges are located in the HormR domain and two more are found in the GAIN subdomain B. **B** The GPS is solvent accessible at the bottom of a groove. A cacodylate ion from the crystallization buffer binds in the groove. **C** Close-up view of the GPS. The (2F$_o$-F$_c$)-type electron density of the GPS residues is shown at a contour level of 0.7 σ$_{rms}$. The sidechain of H910 has been refined in two alternative conformations with occupancies of 0.57 (cyan) and 0.43 (magenta). **D** Comparison of the crystal structures of the HormR and GAIN domains of B2 (in color) and B3 (grey, PDB id 4dlo[4]) **E** Comparison of the GPS structures of B2 and B3. The GAIN domain structures have been superimposed based on subdomain B. **F** Superposition of the crystal structures of B2, L1, G1, G3, G6, and F1. The thickness of the tube indicates the magnitude of the respective crystallographic B-factors, which increase mostly with conformational flexibility in the crystal. The view focuses on the GPS and the flanking flaps. All superimposed structures were aligned to the GAIN subdomain B. GAIN GPCR autoproteolysis inducing, HormR Hormone receptor.

(Supplementary Table 1), and the crystal structure was determined to 2.2 Å resolution (Fig. 2 and Supplementary Fig. 4; Supplementary Table 2).

The refined model comprises residues 532 to 921. Two loops of residues 605-609 and 770-810 are disordered and not resolved. In agreement with gel electrophoresis experiments (Supplementary Fig. 2), the electron density showed that no cleavage had occurred at the GPS. The GPS is located at the bottom of a groove, which is flanked by residues 848 to 856 (flap 1) on one side and 879 to 886 (flap 2) on the other side (Fig. 2B and Supplementary Fig. 5). These flaps have been shown to be flexible in MD simulations of L1, G1 and E5 resulting in open states, in which the GPS is well accessible to the solvent, and closed states, in which in particular flap 1 covers the GPS[18]. Notably, the long, disordered loop region 770-810 of B2 is located next to flap 1 and may contribute to its flexibility. In comparison to other GAIN domain structures, the *h*B2-HG crystal structure represents an open state concerning the flap conformations (Fig. 2B). A cacodylate ion from the crystallization buffer has bound in the groove next to the GPS.

The cleavage site motif of *h*B2 is comprised of H910$^{GPS.-2}$, L911$^{GPS.-1}$ and S912$^{GPS.+1}$ (superscript indicating the position relative to the cleavage site, in accordance with GAIN domain generic residue numbers)[28]. H910 is present in two alternative conformations (Supplementary Fig. 6). In the main conformation, which is shown in Fig. 2C, the imidazole group is oriented closer to the nucleophilic S912 side chain. For a direct hydrogen bonding interaction between S912 and H910, which is required for proton transfer as the assumed first step of GPS

hydrolysis (Fig. 1), S912 needs to change conformation from the t-rotamer observed in the crystal structure to the m-rotamer, and the imidazole group of H910 needs to flip. The side-chain flip of H910 in the crystal structure was modeled based on a hydrogen bonding interaction of the N$_\epsilon$-atom with a nearby water molecule. Thus, neither the S912 nucleophile nor the H910 base is positioned to initiate the cleavage reaction in the crystal structure. However, both residues are expected to be able to change rotamers, as there are no steric clashes preventing these conformational changes. If the rotamers are manually changed to the energy minimum of a serine m-rotamer (χ$_1$ = −65.0°) and to a flipped H910 side chain, the H910 N$_\delta$-atom and the S912 O$_\gamma$ are 2.8 Å apart and well positioned for a hydrogen bonding interaction.

A comparison of the HormR-GAIN domain structures of B2 and B3 shows a similar relative orientation of the two domains (Fig. 2D). The disordered loop of subdomain B is much shorter in the B3 structure, and only two residues were not modeled due to flexibility. There are no significant differences in the overall fold of the two related GAIN domains, which share 49.4 % sequence identity and superimpose with a root-mean-square deviation (RMSD) of 2.22 Å for all common 340 C$_\alpha$-atoms and 1.38 Å for 298 C$_\alpha$ atoms (omitting some deviating loop conformations). The secondary structure elements (Fig. 2D) and the GPS triad (Fig. 2E) superimpose fairly well, but interestingly the largest differences in the main chain structures are observed in the regions surrounding the GPS (Supplementary Fig. 5). A comparison to the crystal structures of cleaved GAIN domains shows

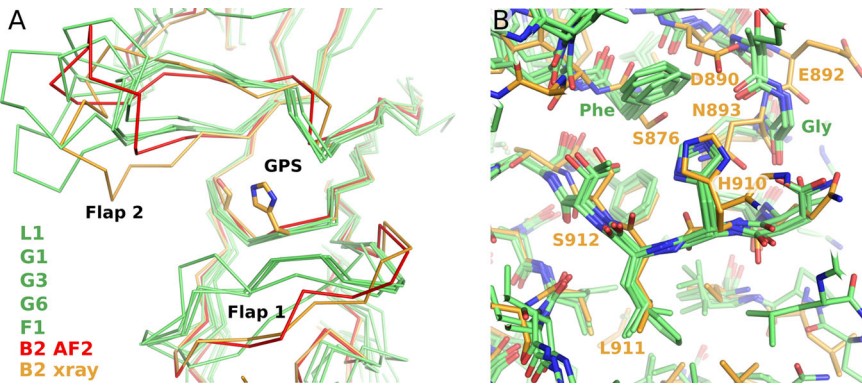

**Fig. 3 | Comparison of the GPS and its environment between the B2 GAIN domain and cleavage-competent receptors. A** Superposition of the $C_\alpha$-traces of the B2 GAIN domain crystal structure (orange, "B2 xray") with its AlphaFold2 model (red, "B2 AF2") and the AlphaFold2 models of the uncleaved GAIN domains of L1, G1, G3 and G6 (all green), for which crystal structures of cleaved GAIN domains have been determined. **B** Comparison of the GPS and its environment between these GAIN domain models showing the conserved phenylalanine (label "Phe") and glycine (label "Gly"). The B2 AF2 model is omitted here. GPS GPCR proteolysis site.

that flap 1 is closed in L1, G1, G3, and G6 whereas in F1 it is in a similar open conformation as in B2 (Fig. 2F). This comparison also demonstrates the conformational diversity of the two flaps and of the loop next to flap 1. In addition, these regions often display increased B-values resulting from some flexibility in the crystal (Fig. 2F). In the B3 GAIN domain structure, serine at the GPS.+1 position (the position directly after the cleavage site) is an outlier in the Ramachandran plot in both chains observed in this crystal structure[4]. Similarly, a strained main chain conformation for serine at the GPS + 1 position is also observed in the B2 GAIN domain with main chain torsion angles of $\psi = -159.5°$ and $\varphi = -179.1°$.

### The GAIN domain of ADGRB2 does not show efficient GPS cleavage even at elevated temperature or in the presence of hydroxylamine

Following expression in HEK293S GnTI⁻ cells, the *h*B2-HG construct was always detectable via its C-terminal twin Strep-Tag and no traces of cleaved NTFs were found throughout the purification process. As the crystal structure analysis demonstrated a properly folded GAIN domain and GPS structure, we tested whether thermal activation might overcome an activation barrier of the autoproteolysis reaction. Incubation of the protein for up to one hour at 60 °C did not result in GPS cleavage (Supplementary Fig. 7). For other systems of *cis*-auto-cleavage, it has been observed that the reaction can be stalled at the ester intermediate state[29]. Although an ester intermediate was not observed in the crystal structure, there might be an equilibrium between the highly populated initial peptide ground state and a higher-energy ester intermediate, which cannot proceed via the final hydrolysis step. Such ester intermediate states can be hydrolyzed by added hydroxylamine[29]. We added up to 500 mM hydroxylamine to the purified *h*B2-HG construct; no GPS cleavage was observed (Supplementary Fig. 7).

### Analysis of possible factors influencing GAIN domain autoproteolytic activity

To understand the structural determinants that might contribute to the apparent autoproteolysis resistance of the B2 GAIN domain, we compared it to cleavage-competent aGPCRs. Because structures of pre-cleavage states of GAIN domains with the H-X-S/T motif are not available, we resorted to a comparison to AF2 models. We generated models for the GAIN domains of receptors G1, G3, G6, and L1, for which crystal structures of GAIN domains in the cleaved state are available and for which the cleavage competence has been demonstrated for the full-length receptors (G1[30], G3[31], G6[32], and L1[4,12]).

A comparison of the *h*B2-HG crystallographic and AF2 models with the AF2 models of the pre-cleavage states of B2, L1, G1, G3, and G6 indicates minor differences in the conformation of the main chain of the GPS region in addition to the different conformations of the flap regions (Fig. 3A). A striking difference in the immediate environment of the GPS concerns residues S876 and N893 as well as the main chain fold of the region around N893 (Fig. 3B). S876 (GAIN generic residue number S10.49[28]) is substituted by a phenylalanine in 21 of the human GAIN domains, including the autoproteolysis-competent receptors compared in Fig. 3B, but also in other receptors, for which GPS-autoproteolytic activity has been detected in previous work (Supplementary Fig. 8). The side chain of this phenylalanine (or tyrosine in E3) interacts with the histidine base of the GPS in a T-shaped π−π interaction. N893 is unique in B2 and it interacts with the H910 side chain via π-stacking and with hydrogen bonds to the main chain atoms (Fig. 2C). This residue corresponds to a strictly conserved glycine in the autoproteolysis-competent human receptors and the glycine is part of a region near the GPS that shows little differences in the AF2 models (Fig. 3B and Supplementary Fig. 8A). The next residue is C894[S12.50], which is part of a disulfide bridge that is present in all human aGPCR GAIN domains.

Considering the mechanistic model of GPS autoproteolysis[14] (Fig. 1), the following factors may contribute to the catalytic efficiency of the cleavage reaction: (i) the protonation state of the histidine base, (ii) the relative orientation of the alcohol (Thr or Ser) nucleophile and the histidine sidechain for efficient proton transfer, (iii) the position of the hydroxyl group for nucleophilic attack on the carbonyl carbon of the adjacent peptide bond, (iv) strain of the peptide bond (i-iv in the uncleaved state 1), (v) stabilization of the tetrahedral transition state or intermediate (2), (vi) protonation of the leaving amine group for efficient conversion to the ester intermediate (3), (vii) position and activation of a water nucleophile for attack on the carbonyl group of the ester intermediate, (viii) protonation of the alkoxide leaving group of the threonine or serine side chain of the CTF (4) in the cleaved protein, and (ix) stabilization of the tetrahedral intermediate formed during ester hydrolysis.

In the AF2 models of L1, G1, G3, and G6, the histidine base was present in the p80-rotamer, where the $N_\delta$-nitrogen is facing towards the Ser/Thr nucleophile. For G6, the threonine nucleophile was not only predicted in the p-conformation but also at a distance of 3.3 Å, thus positioned to interact with the histidine base for proton transfer. However, similar to the situation described above for the B2 GAIN domain crystal structure, the Ser/Thr nucleophiles are free to adopt other rotamers.

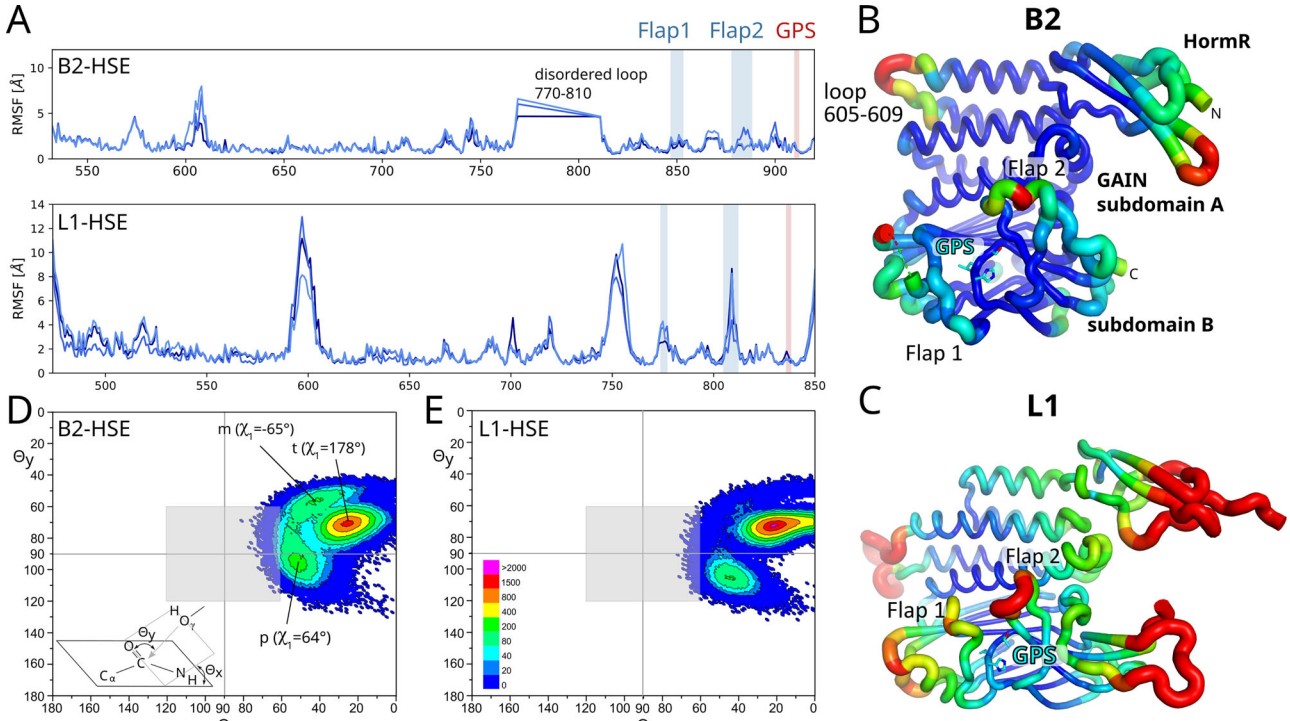

**Fig. 4 | Flexibility of residues during MD simulations. A** Plot of the root mean square fluctuations (RMSF) of the $C_\alpha$-coordinates of individual residues of B2 (top) and L1 (bottom) for the three replicas of the HSE simulations (dark blue: r1, medium blue: r2, bright blue: r3). **B** RMSF values of the $C_\alpha$-coordinates mapped onto the fold of the B2 HormR-GAIN domains such that high mobility is indicated by red color and thick tube regions and low mobility by blue color and thin tube regions. The RMSF values were calculated for all three replicas of the B2-HSE simulations. **C** RMSF values of the three replicas of the L1 HSE simulation mapped onto the fold of the L1 HormR-GAIN domains. Scatterplots for the orientation of the hydroxyl nucleophile (S912 for B2 and T838 for L1) relative to the carbonyl carbon atom of the scissile peptide bond in MD simulations of the B2 and L1 GAIN domains. The definition of angles $\Theta_x$ and $\Theta_y$ is shown in the scheme on the left and was adapted from Radisky and Koshland[33]. **D** Scatterplot for a total of 452021 frames of the three replicas of the B2 simulations with H910 protonated at $N_\varepsilon$. **E** Scatterplot for a total of 452124 frames of the three replicas of the L1 simulations with H836 protonated at $N_\varepsilon$ (10 ps steps of 1500 ns trajectories). The grey square indicates the arbitrary area of $\Theta_x = 90° \pm 30°$ and $\Theta_y = 90° \pm 30°$ used to quantify the number of frames with a favorable geometry of the hydroxyl group for nucleophilic attack on the carbonyl carbon in Table 1. The scale indicates the color coding for both plots to visualize the number of frames for which a given pair of $\Theta_x$ and $\Theta_y$ (binned in 1° steps) is observed. GAIN GPCR autoproteolysis inducing, HormR Hormone receptor, HSE Histidin side chain protonated at $N_\varepsilon$.

## MD simulations of the ADGRB2 and ADGRL1 GAIN domains

We performed MD simulations on the GAIN domain of B2 to examine populations with favorable geometry requirements for efficient attack of the nucleophile on the carbonyl group of the adjacent peptide bond as the initial step of the GPS cleavage mechanism. In addition, the computational estimation of $pK_a$ values strongly depends on details of the residue environment and is therefore preferably assessed for an ensemble of available conformations observed in MD simulations. For a comparison of the B2 MD simulations to those of a cleavage-competent receptor, we used the uncleaved wild-type L1 GAIN domain. Cleavage-competent conformations of the GAIN domain may differ from low-energy states observed in crystal structures or in energy-minimized structural models. We therefore employed MD simulations to analyze and compare the conformational landscapes of the L1 and B2 GAIN domains for factors that may influence the initial step of GPS autoproteolysis. A model for the uncleaved rat L1 GAIN domain was generated based on the crystal structure of the cleaved L1 GAIN domain[4]. The histidine base within the GPS (L1: H836; B2: H910) must be in one of two deprotonated neutral states to act as a general base. We tested the behavior of the L1 and B2 receptor, each in three protonation states: HSE (neutral with proton at $N_\varepsilon$), HSD (neutral with proton at $N_\delta$), and HSP (both nitrogens protonated), in three independent 1500 ns replica simulations. The backbone-RMSD values reached a plateau after about 100-200 ns, indicative of a stable conformation (Supplementary Fig. 9). Larger RMSD values of ~6-7 Å were observed for L1 compared to B2 (~2-3 Å). This is primarily due to a rigid-

body motion of the HormR domain against the GAIN domain. Additionally, we observe much higher flexibility of loop regions in L1, reflected by higher root mean square fluctuations (RMSF) of the $C_\alpha$-coordinates (Fig. 4A–C). High mobility is observed in particular for flap 2. The loop between residues 605 and 609 also shows high mobility in agreement with its missing electron density in the crystal structure. An analysis of the predicted $pK_a$ values of H910 in B2 and H836 in L1 reveals that the side chains exist in a dynamic continuum of protonation states, favoring the HSE state (Supplementary Fig. 10).

Radisky and Koshland[33] analyzed the position of the attacking hydroxyl nucleophile in complexes of serine proteases with peptide or protein inhibitors, which presumably can react to form acyl-enzyme intermediates in equilibrium with the Michaelis complex but are not completely hydrolyzed due to a slow deacylation step. In 79 structures, the nucleophile positions were tightly clustered at angles of about $\Theta_x = 90°$ and $\Theta_y = 90°$ perpendicular to the peptide plane at the carbonyl carbon position. The optimal angle for attack of a nucleophile on a carbonyl group, also known as the Bürgi-Dunitz angle, was first described as $\Theta_y = ~105°$ based on theoretical calculations[34] and crystal structures[35]. However, a more recent analysis including a much larger number of crystal structures determined $\Theta_y = 90°$ as the minimum energy angle[36]. A comparison of the scatterplots depicting the positions of the alcohol nucleophile relative to the carbonyl group for the simulations of B2 and L1 (Fig. 4D, E) shows that in both simulations, the nucleophile is mostly located at $\Theta_x \approx 25°$ and $\Theta_y \approx 72°$. In the simulation of L1, however, the nucleophile less often deviates from this

**Table 1 | Geometric parameters describing the orientation of the S/T alcohol nucleophile, the histidine base and the scissile peptide bond in the MD simulations of the B2 and L1 GAIN domains**

| | $\Theta_x = 90° \pm 30°$ and $\Theta_y = 90° \pm 30°$ | Hbond[a] | Hbond and $\Theta_{x/y} = 90° \pm 30°$ | $\chi_1$[b] His$^{GPS.-1}$ | | | $\chi_1$[b] Ser/Thr$^{GPS.+1}$ | | |
|---|---|---|---|---|---|---|---|---|---|
| | | | | +60° | −60° | −180° | +60° | −60° | −180° |
| B2 r1 | 3597 | 19 | 1 | 1.9% | 5.2% | 69.3% | 26.2% | 19.8% | 45.6% |
| B2 r2 | 1020 | 9 | 0 | 4.2% | 17.2% | 50.3% | 4.9% | 17.3% | 67.6% |
| B2 r3 | 2876 | 35 | 0 | 1.1% | 14.0% | 63.6% | 21.9% | 14.1% | 55.7% |
| L1 r1 | 147 | 1807 | 23 | 60.2% | 33.9% | 2.7% | 4.8% | 0.7% | 60.5% |
| L1 r2 | 35 | 1351 | 3 | 98.1% | 0.7% | 0.0% | 2.6% | 1.7% | 59.4% |
| L1 r3 | 359 | 7469 | 33 | 95.8% | 1.8% | 0.7% | 16.0% | 2.9% | 53.3% |
| B2FG r1 | 517 | 268 | 16 | 10.2% | 53.9% | 25.2% | 4.5% | 6.3% | 64.5% |
| B2FG r2 | 537 | 538 | 14 | 7.3% | 57.5% | 25.0% | 1.5% | 5.8% | 69.7% |
| B2FG r3 | 51 | 1 | 0 | 1.6% | 77.8% | 14.7% | 0.1% | 3.1% | 70.8% |

The values specify the number or percentage of frames in which the specified condition is fulfilled. These simulations were carried out with a neutral histidine base protonated at $N_\epsilon$ (HSE). Results from simulations with protonation at $N_\delta$ are shown in Supplementary Table 3. The underlined $\chi_1$ values denote rotamers, in which a hydrogen bonding interaction between the histidine and the S/T alcohol nucleophile is possible. B2FG denotes the S876F/N893G B2 variant. Heatmaps for the frequency of hydrogen bonding between the S/T nucleophile and the histidine base are shown in Supplementary Fig. 11. The total number of frames sums up to about 452000 for the sum of the three replicas of each simulation setup.

[a]A favorable hydrogen bonding geometry was defined as a distance of 3.2 Å or less between the histidine nitrogen atom and the Ser/Thr $O_\gamma$ atom and a deviation of less than 30° of the $O_\gamma$-H⋯N angle from linearity.

[b]The $\chi_1$ torsion angle was assigned to the given value if it deviated by less than 30° from this value.

catalytically suboptimal position compared to B2. Consequently, for the B2 GAIN domain, the nucleophile more often approaches a position favorable for nucleophilic attack compared to L1 (Table 1).

In contrast, a hydrogen bonding interaction of the histidine base and the hydroxyl nucleophile is more frequently observed in the simulation of L1 (Table 1 and Supplementary Fig. 11). As a result, the L1 receptor rarely, but much more frequently than B2, adopts a conformation in which the threonine hydroxyl group can transfer its proton to the histidine base and in which it is positioned for nucleophilic attack on the adjacent peptide carbonyl group in a concerted manner (59 frames for L1 vs. only 1 for B2, Table 1). In the L1 simulations with HSE (His $N_\epsilon$ is protonated, Table 1), suitable hydrogen bonding geometries between H836 and T838 are much more frequently observed in comparison to the HSD simulations (Supplementary Table 3). The more frequent occurrence of the H836-T838 hydrogen bond in L1 compared to B2 is due to the high occupancy of the histidine p80 rotamer ($\chi_1 = +60°$) in the simulation (Table 1). In this rotamer, the $N_\delta$ of the histidine base is positioned favorably for hydrogen bonding interaction with the Ser/Thr nucleophile when this adopts m or p rotamers. Whereas the favorable m rotamer of the Ser nucleophile shows even higher occupancy in the B2 simulation (Fig. 4 and Table 1), the low occupancy of the p80 rotamer of H912 correlates with the rare occurrence of the H910-S912 H-bonding contact in B2 and might contribute to the apparent cleavage deficiency of this receptor. We suspected that the different environment of the imidazole side chain of the histidine base in L1 and other cleavage-competent receptors is responsible for ensuring a high occupancy of the p80 rotamer. As noted above, the most prominent differences between cleavage-competent receptors and B2 are the presence of a phenylalanine or tyrosine residue (forming a T-shaped π−π interaction with the histidine base) in cleavage-competent receptors in place of S876$^{S10.49}$ and the presence of a glycine at the position of N893 in B2 (Fig. 3B).

To investigate the influence of the conserved glycine and phenylalanine residues on the conformation of the GPS, we performed another set of MD simulations of an S876F/N893G variant of the B2 HormR-GAIN domains (Table 1). Indeed, in two of the three replicas of 1500 ns, favorable geometries for nucleophilic attack of S912 are observed more frequently than for the wild-type receptor (30 frames for the S876F/N893G variant compared to 1 frame for the wild-type domain). As with the wild-type B2, the frequency of favorable Θ angles remains lower in simulations of L1 compared to the B2 S876F/N893G

variant, but the frequency of hydrogen bonding interactions between H836$^{GPS.-2}$ and T838$^{GPS.+1}$ is higher in the L1 simulations (10627 frames for L1 compared to 807 frames for the B2 S876F/N893G variant). Concerning both criteria, the assessed viability for cleavage of the B2 S876F/N893G variant falls between the wild-type B2 and the L1 GAIN domains.

We inspected the MD frames of L1 simulations with favorable positions of the serine nucleophile and histidine base (Table 1) and noted that in some of these frames a distorted ω angle was marked for the scissile peptide bond, in contrast to all other residues in the direct GPS environment. A systematic analysis of the ω torsion angle $C_\alpha$-C-N-$C_\alpha$, which describes the planarity of the peptide bond, showed that strained conformations of the scissile bond (quantified by having more than 25° deviation from planarity) were observed in 17 (28.8 %) of the 59 "GPS-active" conformations, the latter of which contain the H-bond and $\Theta_{x/y} = 90° \pm 30°$ for the L1 GAIN domain (Table 1). This compares to 1.2 % strained conformations of this peptide bond observed in all frames of the L1 MD simulations. Thus, strained conformations are strongly enriched when the His and Ser residues of the GPS are in a reactive position. A plot of the distortion of the ω angle against the deviation of the position of the Ser nucleophile from the 90° angle optimal for attack on the peptide carbonyl group shows that favorable orientations indeed correlate with strained peptide bonds (Supplementary Fig. 12). This strain of the ground state structure favors nucleophilic attack on the peptide carbonyl group[37–39]. The scissile peptide bond is, however, not unique in comparison to other residues of the GAIN domain concerning the deviations of the ω angle from planarity (Supplementary Fig. 12C). As noted before, the main chain conformation of the Ser/Thr residues is also strained concerning the φ,ψ-angles observed in the B2 and B3[4] GAIN domain crystal structures. In the energy-minimized AF2 model of uncleaved L1, the nucleophilic T838 is also an outlier in the Ramachandran plot. It appears likely that the strained main-chain conformation at T838 contributes to the formation of strain on the scissile peptide bond to favor autoproteolysis.

Taken together, the typical cleavage-competent GAIN conformation is characterized by the ability of the $N_\delta$-atom of the GPS histidine in the p80 rotamer to deprotonate the S/T alcohol nucleophile (Fig. 5A). Nucleophilic addition to the neighboring peptide carbonyl group is supported by strain on the peptide bond, which is strongly correlated with the formation of a favorable orientation of the alcohol nucleophile for nucleophilic attack. The p80 rotamer is stabilized by a

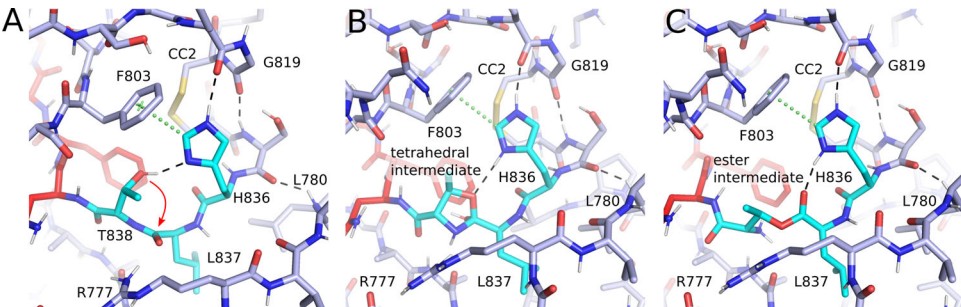

**Fig. 5 | Structural models of the N-O acyl shift in GPS hydrolysis. A** Model of the GPS in a favorable geometry for deprotonation of T838 by the H836 base and nucleophilic attack of the hydroxylate on the carbonyl carbon. The model was obtained from a representative frame of the MD simulations of the uncleaved L1 GAIN domain. H836 adopts the p80 rotamer ($\chi_I = 61°$) and T838 the p rotamer ($\chi_I = 59°$). **B** Model for the tetrahedral intermediate **2** (see Fig. 1). **C** Model of the ester intermediate. The models were generated using the software MOE[101].

T-shaped π–π interaction with a phenyl side chain (F803 in L1) and a hydrogen bond to the carbonyl oxygen of the peptide bond between the highly conserved glycine-cysteine residues N-terminal to cysteine bridge 2 (CC2, Fig. 5A). Further hydrogen bonds of the main-chain of this region upstream of the GPS additionally stabilize the GPS-active state in this model. These interactions were observed in all MD conformations of the L1 simulation, which are considered active concerning the first step of the autoproteolysis reaction.

### Further steps of the autocatalytic GPS hydrolysis mechanism: The histidine of the GPS is unlikely to protonate the amine leaving group in the N-O acyl shift

To analyze interactions that might promote further steps of the autoproteolytic mechanism, we generated a model of the tetrahedral intermediate or transition state of L1 formed after attack of the alcohol nucleophile (Fig. 5B). In this model the protonated catalytic histidine might stabilize the negatively charged intermediate by hydrogen bonding interactions to the oxygen atom that results from the carbonyl oxygen of the peptide bond (2 in Fig. 1). Notably, we did not observe other direct interaction partners positioned to function as an oxyanion hole as known from serine proteases[40]. Based on this model, we assumed that the main chain cannot rearrange such that the histidine would be positioned to directly protonate the leaving nitrogen atom in the next catalytic step. Since no other amino acid residue of the conserved GAIN domain core (Figs. 3B; 5A) is positioned for this protonation step, a solvent water molecule may provide a proton upon expulsion of the nitrogen atom. An in-depth study carried out in parallel to this work investigated the complete GAIN autocleavage mechanism employing quantum-mechanical computational methods[41]. This approach confirmed a steric hindrance of the histidine base to transfer the proton.

After the nitrogen atom has left, the ester intermediate is formed (Fig. 5C). Besides the histidine base, there is no obvious residue in the immediate environment of the GPS that could support hydrolysis of the ester intermediate by stabilization of the transition state (other than R777 of the flexible flap1). The basic primary amine resulting from the released peptide nitrogen might act as a general base to deprotonate a water molecule for nucleophilic attack on the ester carbonyl as the last step of the autoproteolytic mechanism. Alternatively, a conserved glutamate residue (E656 in L1) might deprotonate the water nucleophile[41].

### Cleavage competence of ADGRB2 and ADGRB3 can be restored by mutations in the GPS sequence and its immediate environment

To test the influence of the immediate environment of the GPS sequence on the autocleavage activity, we analyzed the cleavage competence of variants of L1, B2 and B3 using the full-length receptors and isolated HormR/GAIN domains. We achieved significant activation of ADGRB3 for self-cleavage (Fig. 6E, F). This receptor features an RLS motif at the GPS (with R855 at the base position), and the conserved aromatic residue that forms a T-shaped π–π interaction with the histidine base in cleavage-competent receptors (Fig. 6A, D) is L821 in B3. Whereas no cleavage was observed for the full-length wild-type B3 receptor, 6.5 % of the R855H variant with a canonical HLS sequence were cleaved. This value increased to 24.5 % cleavage for the R855H/L821F double mutant, which restores the general base and its π–π interaction for stabilization of the p80 rotamer (Fig. 6E). L821W and L821Y also supported increased cleavage activity via the π–π interaction. Vice versa, the naturally cleavage-competent L1 receptor shows a reduction in its autoproteolytic activity if F803 (the residue for the π–π interaction with the histidine base H836) is replaced by a leucine or alanine, whereas full cleavage competence is retained for F803W and F803Y mutations. These findings demonstrate the importance of the T-shaped π–π interaction of the histidine base for self-cleavage activity. We also tested the influence of corresponding mutations on the full-length B2 receptor (Supplementary Fig. 14). GPS autocleavage was not observed in these experiments, most likely due to the formation of only misfolded protein as indicated by the lack of complex N-glycosylation of the expressed protein.

The cleavage assays of the full-length receptors were performed ~24 h post-transfection. We also measured the percentage of cleaved GAIN domains for the expression of HormR/GAIN domains of the three receptors (L1, B2 and B2, Fig. 7). This enabled us to detect even very low cleavage activity as the reaction could be monitored over prolonged time periods for the purified ectodomain constructs. The cleavage assays for the B3 and L1 ectodomain constructs were in good agreement with the results obtained for the full-length receptors. For the B2 receptor, cleavage activity could also be characterized (Fig. 7). Interestingly, these assays revealed low cleavage activity with ~2–4 % initial cleavage for the wild-type *h*B2-HG construct (termed "wt" in Fig. 7) and the S876F/D890S/E892Q/N893G variant ("basic", Fig. 7B/C). The latter construct contains D890S and E892Q exchanges in addition to the already described S876F and N893G mutations. D890 is a serine in L1, G1, and G3 and an asparagine in G6 and F1. It is located behind the phenylalanine of the His base π–π interaction. E892 is pointing into the solvent and is quite variable in the cleavage-competent receptors. These two mutations were introduced to further mimic the GPS environment in the cleavage-competent receptor L1.

### The flaps around the GPS contribute to full autocleavage activity

The residual cleavage competence of the F803A and F803L variants of L1, as well as the limited activity of the B2 S876F/D890S/E892Q/N893G and B3 R855H/L821F variants, indicates that additional factors further

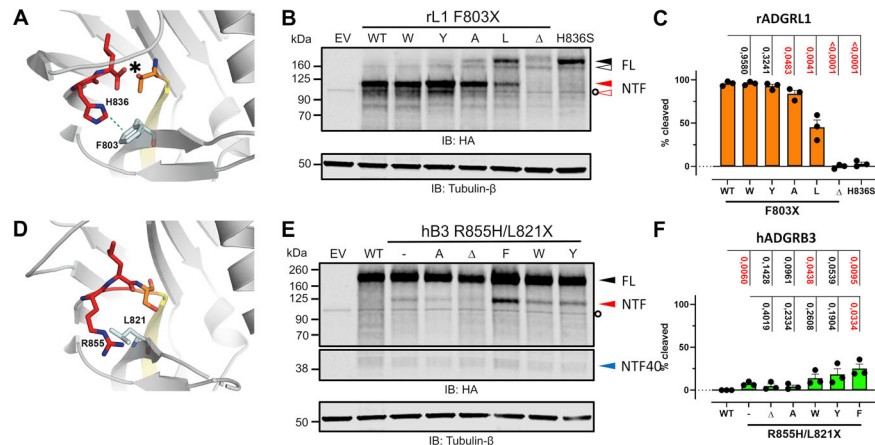

**Fig. 6 | Influence of the π–π interaction between the His^GPS,−2 base and surrounding residues on GAIN domain autocleavage of the full-length L1 and B3 receptors. A** π–π interaction between H836 and F803 observed in the crystal structure of rat L1 (pdb id 4dlq)[4]. These residues correspond to R855 and L821, respectively, in *h*B3 (**D**, crystal structure 4dlo[4]). HEK293T cells were transiently transfected with constructs containing mutations causing either a disruption of π–π interaction in *r*L1 (**B**, **C**) or a reintroduction of π–π interaction in *h*B3 (**E**, **F**). GPS cleavage of the mutants was analyzed by WB targeting the N-terminal HA tag of the receptors, using 30 µl of cell lysates. Tubulin served as a loading control. Bands representing uncleaved full-length receptor (FL) and N-terminal fragment (NTF) are indicated in black and red triangles, respectively. Fully (N- and O-glycosylation) and

immaturely (no O-glycosylation) glycosylated populations of rat L1 are indicated in closed and open triangles, respectively. "NTF40" denotes an N-terminal fragment of ~43 kDa apparent molecular mass indicating additional processing of *h*B3, similar to *h*B1[102] and *h*B2 (Fig. 6A). For further details including calculated molecular masses, deglycosylation experiments and interpretations see Supplementary Fig. 13. Quantifications of the extents of GPS cleavage of the mutants in (**B**), (**E**) are shown in (**C**), (**F**), respectively. Data are presented as mean values ± SEM with *N* = 3 independent experiments. The extent of GPS cleavage is calculated by dividing the intensity of NTF by the sum of the intensities of NTF and FL. EV Empty vector, HA Hemagglutinin, IB Immunoblot, WT Wild type. See the "Methods" section for statistics.

contribute to cleavage competence. As structural comparisons of the B2, B3 and L1 receptors revealed no significant differences in the environment of the GPS other than the mobile and diverse flaps, we decided to characterize the autocleavage activity of chimera generated by flap exchanges between the B2 and L1 receptors (Supplementary Fig. 15A). As some residues of the flaps are interacting with neighboring regions of the GAIN subdomains A or B, we chose the exchanged regions such that mostly weakly conserved, solvent-exposed (noninteracting) residues were swapped. Two mutations also involved residues of the last α-helix of subdomain A (helix 6)[28], since these are in contact with flap 2. In addition to flaps 1 and 2, we also prepared flap exchange chimera of neighboring loop regions (denoted as flap 1B and flap 1 C), which showed elevated B-factors in crystal structures (Fig. 2F) and elevated RMSF values in the MD simulations (Fig. 4C). Implementation of the L1 flaps into the B2 GAIN domain significantly increased the autocleavage activity of the secreted HormR-GAIN domains (Fig. 7C). The fractions of cleaved B2 receptor with flap exchanges are between the values observed for the F803A or F803Y mutations of the *r*L1-HG construct (Fig. 7C). Vice versa, the presence of flap 2 of the B2 receptor significantly diminished the autocleavage activity of L1 for the isolated HormR-GAIN construct and the exchange of both flaps resulted in an autocleavage deficient variant (Fig. 7C). Whereas the combined exchange of flaps 1 and 2 resulted almost in a complete loss of autocleavage activity for L1, an additive effect of the exchange of flaps 1 and 2 was not observed for B2. Experiments to detect the influence of the flap exchanges in full-length constructs of the B2 receptor failed due to protein misfolding (Supplementary Fig. 16).

### For slow-cleaving variants, the rate constant of the autocleavage reaction can be determined

Purification of slow-cleaving variants of the HormR-GAIN domain constructs of B2, B3 and L1 allows for a determination of the rate constant of the autoproteolysis reaction in the purified proteins. The rate constants of wild-type *h*B2-HG and the S876F/D890S/E892Q/N893G variant are in good agreement with the results of the cleavage assays concerning the low cleavage activity and the absence of a large

influence of the π–π interaction for this receptor (Fig. 7D, E and Supplementary Fig. 15). These assays demonstrate that properly folded GAIN domains can undergo autocleavage outside of the secretory pathway. However, for some constructs, the initial cleavage fraction after protein purification (timepoint 0 in the graphs in Fig. 7E) is quite high compared to the cleavage rate observed in the time course of the following 14 days. The cleavage rate of the *h*B3 variants R855H and R855H/L821Y or hB2 wt and S876F/D890S/E892Q/N893G ("basic") is of comparable magnitude before and after protein purification. For these proteins, the fitted curve extrapolates to ~0 % cleaved protein at the timepoint of transfection (−2 d), which is 48 h before the first sample was taken after elution of the purified protein from the Ni-NTA beads. In contrast, for constructs *h*B2 F1BC (exchange of flap 1 and the two neighboring loops, see figure legend Fig. 7 for the abbreviations), *h*B2 F1, *r*L1 F803Y or *r*L1 F803W, the fraction of cleaved protein is extrapolated to large values at the point of transfection indicating that the cleavage rate was much higher before the elution step. Furthermore, this initial cleavage sometimes differed substantially between independent expressions, as illustrated by the large differences in initial cleavage in Fig.7C or the time course of GPS cleavage for two expressions of the *h*B2 F1 variant, denoted as "*h*B2 F1 high" and "*h*B2 F1 low" for high and low initial cleavage. The cleavage rate of the isolated proteins in these two experiments is identical within the accuracy achieved in these experiments. It is also noteworthy that the large fraction of cleaved protein observed for the *r*L1 variants F803W and F803Y is in contrast to their relatively slow cleavage rates of the isolated proteins in comparison to the *h*B2 flap exchange variants or *h*B3 R855H/L821Y, which cleave much faster as isolated proteins but have a lower initial cleavage fraction. This difference in the cleavage rates might be due to accelerated cleavage in the secretory pathway or due to differences in the buffer conditions of the expression medium and the elution buffer of the Ni-NTA purification step.

### Evolutionary conservation of the B2 GAIN domain
We analyzed the conservation of the amino acid sequence of the human B2 receptor by a comparison to 148 mammalian orthologs to indicate functional roles of specific protein residues for folding,

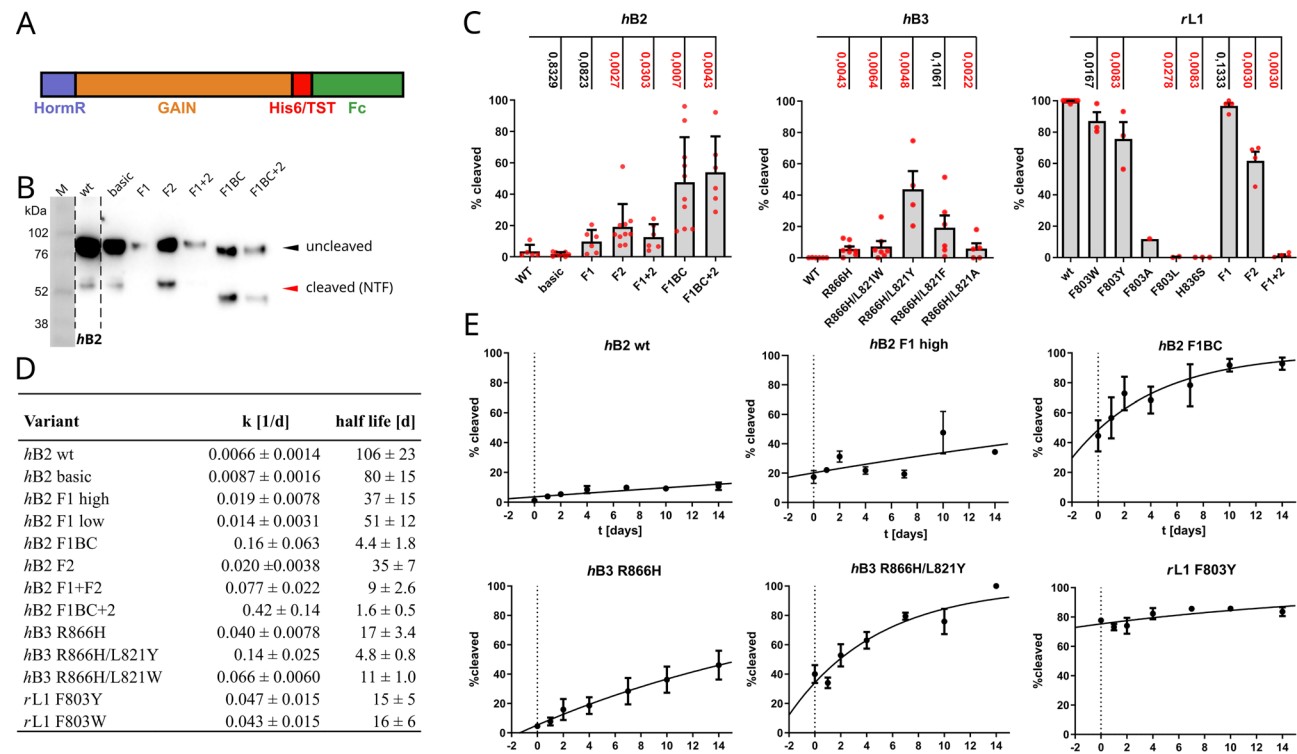

**Fig. 7 | Autocleavage assays of HormR-GAIN domain constructs of _h_B2, _h_B3 and _r_L1. A** Scheme of the constructs used in the cleavage assays. **B** Western blot results of cleavage assays of different _h_B2 constructs. "wt" denotes the wild-type sequence, "basic" the S876F/D890S/E892Q/N893G variant, the other variants contain exchanges of the _h_B2 flap regions with _r_L1 flap regions as shown in Supplementary Fig. 15A. The NTFs containing an exchange of the long flap 1B of _h_B2 against the corresponding shorter flap of _r_L1 run at lower apparent molecular mass. **C** Results of the cleavage assays for selected variants of the three receptor ectodomains. The cleavage fractions were determined from western blots of an SDS-PAGE analysis of protein purified using Ni-NTA beads, showing the cleavage state 2 days after transfection. For _h_B2, at least five biological replicates were used. For hB3 and _r_L1, at least four biological replicates were set up. Data points were discarded if the sum of the intensities of cleaved and uncleaved bands on the western blot was too low or if the bands and background were affected by obvious artefacts. Data are presented

as mean values ± SEM. **D** Results of kinetic analysis of GPS autoproteolysis for selected variants. **E** Development of the cleavage fraction determined via western blots at different time points (0, 1, 2, 4, 7, 10 and 14 days) after purification of the ectodomains. Plots for all variants are shown in Supplementary Fig. 15 together with the number of replicas. The flap exchange chimeras have been referred to by the following abbreviations. F1: flap 1, F1BC: flap 1 + 1B + 1 C, F2: flap 2, F1 + 2: flap 1 + 2, F1BC + 2: flap 1 + 1B + 1 C + flap 2. "hB2 F1 high" and "hB2 F1 low"(Supplementary Fig. 15) are two experiments using the same variant and conditions, but the initial amount of cleavage differs so much that the data have been evaluated independently. Flaps have been exchanged between the _h_B2 and _r_L1 receptors. Data are presented as mean values ± SD. Fc Fragment crystallizable, GAIN GPCR autoproteolysis inducing, HormR Hormone receptor, TST Twin-Strep-tag. See the "Methods" section for statistics.

stability and in particular for GPS hydrolysis (Fig. 8). Flaps 1 and 2, as well as the long, disordered loop of residues 770-810 (Flap 1B, Supplementary Fig. 17) show little sequence conservation. We did not find any residues in the immediate environment of the GPS that are highly conserved to suggest a catalytic function, with the exception of E699[H6.50], which is well conserved as a glutamate or aspartate in aGPCRs[28]. Notably, the histidine base position (H910) shows significant variance. In detail, the GPS.−2 position contains His (31.6 %), Arg (30.8 %), Gln (28.0 %), Lys (8.4 %), Tyr (0.6 %), Pro (0.5 %), and Ser (0.1 %) residues. The GPS.+1 residue contains a serine/threonine in 81.4 % and an alanine in 18.6% of the sequences. Thus, the majority of the B2 orthologs do not contain the histidine base at the GPS required for high cleavage activity. In L1, the GPS and also the neighboring flap 1 region is much higher conserved compared to B2 and B3 (Fig. 8). However, the R[GPS.−2] residue at the GPS of B3 is also strictly conserved with one exception being the fish _Tetraodon nigroviridis_, although this residue has no obvious function for GPS cleavage or _Stachel_ dissociation. It may have a structural role in the formation of a salt bridge with a strictly conserved aspartate residue (D823 in _h_B3).

## Discussion
Kinetic studies of autoproteolysis of the _h_B2 receptor ectodomain revealed a very low cleavage activity with a half-life of around 100 days,

despite the presence of a canonical HLS motif. For the L1 receptor, the F803Y or F803W mutations affecting the π−π interaction, which probably do not drastically reduce autocleavage activity, result in a half-life of the receptor of several days (for the purified ectodomain). Thus, even for this presumably cleavage-efficient receptor, autoproteolysis is slow compared to endopeptidases that cleave peptide bonds with $k_{cat}$ values of ~10 s[−1] for bovine trypsin[42]. We crystallized a construct consisting of the HormR and GAIN domains of _h_B2 and determined the crystal structure in the uncleaved state. After structure determination of the HormR-GAIN construct of _h_B3[4], this is the second high-resolution structure of an uncleaved aGPCR GAIN domain, whereas six structures are available for GPS-cleaved GAIN domains (L1, L3, F1, G1, G3, G6)[4–8,10]. A comparison of the B2 and B3 GAIN domains with AF2 models of cleavage-efficient aGPCRs indicated that the fold of the GPS sequence is rather similar and that the immediate environment of the GPS in GPS-active receptors does not contain any conserved residues besides the GPS sequence itself that are capable of acid-base catalysis, which would be a typical situation for a proteolytic enzyme. On the contrary, the GPS environment lacks such residues and displays a surprisingly diverse environment, particularly in the form of two flaps, which are flexible and flank the GPS groove, as noted in previous work[18].

However, there are also structural features that are conserved in GAIN domains with high cleavage activity. They contain a highly

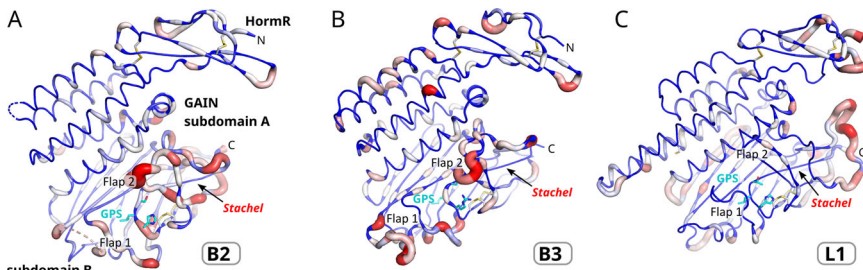

**Fig. 8 | Conservation of the B2, B3, and L1 HormR and GAIN domains.** Sequences of the HormR and GAIN domains of **A** 148 (B2 domains), **B** 137 (B3 domains), and **C** 167 (L1 domains) mammalian species were aligned, and the Shannon entropy was calculated for each residue, which quantifies the variability of the sequence positions. Residues of high sequence conservation are shown as blue thin tubes and residues of high sequence variation as thick red tubes. GAIN GPCR autoproteolysis inducing, GPS GPCR proteolysis site, HormR Hormone receptor.

conserved phenylalanine ($F^{S10.49}$) that interacts with the histidine side chain of the canonical H-X-S/T GPS sequence via a T-shaped π–π interaction. Together with a hydrogen bond to the main chain carbonyl group of the residue preceding the highly conserved Gly-Cys motif in GPS-active aGPCRs, this interaction positions the histidine side chain to accept a proton from the serine or threonine nucleophile. Quantum mechanics calculations show that this T-shaped π-π interaction with the C-H of the histidine pointing towards the phenylalanine π-system is a favored geometry[43]. Whereas the interaction of a neutral His with Phe was found to be π–π in nature, the protonated His forms attractive cation–π interactions. The cysteine $C^{S12.50}$ of the Gly-Cys motif forms a disulfide bridge that is present in most aGPCR GAIN domains. MD simulations of the uncleaved, cleavage-competent L1 GAIN domain in comparison to the mostly GPS-inactive B2 GAIN domain underlined the importance of these interactions for the formation of cleavage-competent conformations, in which the histidine base, the serine/threonine nucleophile and the electrophilic carbonyl carbon are in a favorable position for the first step of the catalytic mechanism (Fig. 5A). The formation of this conformation further generates strain on the scissile peptide bond, which likely favors nucleophilic addition of the alcohol to the peptide carbonyl group.

Mutation of the π-stacking phenylalanine to an alanine or leucine markedly reduced the autoproteolytic activity of L1, whereas the cleavage-inefficient B3 GAIN domain was significantly activated by a double mutation that installed the phenylalanine-histidine π-stacking pair at the GPS (Fig. 6). The residual autoproteolysis of L1 mutants lacking π–π interactions of the histidine base and the only partial gain of cleavage efficiency of B3 in the R855H/L821F variant indicate that other factors contribute to maximal autoproteolytic activity. Preparation of flap exchange chimera of the B2 and L1 receptors demonstrated that the flexible and diverse flaps around the GPS are a third requirement for maximal GPS autocleavage activity besides the canonical H-X-S/T GPS sequence and the π–π interaction (Fig. 7). It may appear surprising that mobile loops with low conservation among cleavage-competent receptors contribute to catalysis. Catalytic residues in enzyme active sites, in contrast, are usually well conserved. A likely scenario is that these mobile flaps contribute strain to the scissile peptide bond (see also discussion below on other cis-autoproteolytic proteins). Formation of strain likely depends not on particular amino acid residues but can be achieved by several types of flap loop structures. We analyzed MD trajectories obtained from three 1500 ns-long MD simulations of the flap exchange variants for strained conformations of the scissile peptide bonds, but could not find a correlation between autocleavage activity and the number of frames with strain on the peptide bond. This may be due to the inherent limitations of empirical force fields employed in MD simulations to monitor strained and polarized molecular geometries. Indeed, in a more detailed QM/MM analysis, a strained conformation of the peptide bonds of the catalytic triad is observed, with a minor energetic impact on reactant

enthalpy[41]. Additionally, the strained states correspond to higher energy levels, which necessitate much longer simulation times to sample conformations in which the mobile flaps exert significant additional strain on the scissile peptide bond of the GPS.

*Cis*-autoproteolysis, where a serine, threonine or cysteine nucleophile attacks the penultimate peptide bond, followed by N→(O, S) acyl shift and formation of the ester intermediate, has also been observed in other proteins[44], including N-terminal nucleophile (NtN) hydrolases[45,46], the pantetheine hydrolase ThnT[47], Nup98[48] or the SEA domain[38,39]. In mechanistically related nonhydrolytic proteins, the (thio)ester intermediate is attacked by nucleophiles other than water, such as in inteins[49], hedgehog proteins[50] or pyruvoyl enzymes[51]. Despite the common catalytic features, these proteins employ significantly different additional strategies to catalyze the self-cleavage reaction (Supplementary Table 4). A common structural feature is that the scissile peptide bonds are solvent-exposed and positioned in loop regions next to a β-strand. However, the conformation and environment of the processed peptide region are diverse. A factor that appears to be more prominent in cis-autocleavage compared to serine proteases or other enzymes is the use of strain to destabilize the ground state. Distortions from ideal geometry have been observed for the scissile peptide bond and for adjacent torsion angles for many independent systems (Supplementary Table 4). Strain or conformational energy of adjacent bonds is likely temporarily transferred to the scissile peptide bond by protein dynamics. Distortion of the peptide bond results in a drastic increase in the $pK_a$ of the nitrogen such that it may be protonated even before attack of the nucleophile[37]. The role of strain in cis-autoproteolysis has been especially well studied for SEA domains, resulting in a model for the catalytic mechanism where substantial energy of protein folding is invested to generate a strained, flexible pre-cleavage ground state structure, which is poised for nucleophilic attack in the absence of a catalytic base and oxyanion hole[38,52]. Remarkably, even removal of the hydroxyl nucleophile by mutation to an alanine resulted in residual site-specific cleavage activity. These observations on related proteins support the notion that the strain formed at the GPS and the scissile peptide bond by the fold of the GAIN protein is an important factor for the autocatalytic reaction. The dynamics of the mobile flaps around the GPS and the folding pathway might increase this strain.

The observation that the B2 receptor has low cleavage activity with a half-life of ~100 days (Fig. 7), raises the question if the cleaved state is strictly necessary for *Stachel* release and receptor activation, considering findings that indicate that GPS cleavage might also depend on the cell type[4]. Analysis of the GPS sequences of B2 homologs demonstrated that the majority (68.4 %) of these have no canonical GPS sequence that is required for efficient autoproteolysis. In comparison, the GPS of L1 is strictly conserved as an HLT sequence motif, while B3 homologs contain a highly conserved R-V/L-S motif. It remains an open question if the low autoproteolytic activity of B2 is

necessary for receptor signaling in vivo, or if the uncleaved receptor signals also without GPS cleavage, as it has also been shown for the naturally autoproteolysis-resistant receptors mouse ADGRG5/GPR114[26], ADGRC1/CELSR1[53] and ADGRC3/Celsr3[54].

GPS cleavage of full-length B2 was concluded in a previous study[55]. While the majority of the B2 receptor obtained from recombinant expression in U87-MG cells was not cleaved, fragments were observed in a western blot of this cell line and in mouse hippocampal tissue that correspond in size to CTF fragments. Likewise, such fragments were observed in mouse brain lysates and brain immunoprecipitates with anti-BAI1 and anti-BAI3 antibodies raised against cytosolic protein regions[4]. These fragments obtained from endogenously expressed receptors may correspond to the CTFs generated by GPS cleavage since the differences in expression levels, glycosylation[56,57] and environment in the native tissue compared to expression in HEK293 cells might accelerate autoproteolysis[4]. However, it should also be considered that protein fragments can be expressed in native tissue from alternative promoter regions. For B3, as well as for 67% of all aGPCR genes, mRNA transcripts of CTFs without or with a short N-terminal extension have been observed[58]. For B3, a CTF-sized fragment was detected in a western blot of brain tissue derived from a mouse line carrying a 7-bp deletion in exon 10 of BAI3, which encodes part of the GAIN domain and renders autoproteolysis very unlikely[59].

GPS cleavage influences one of the most important functions of the GAIN domain, the release of the *Stachel* agonist for binding to the 7TM domain. As the Ser/Thr residue of the GPS is the first residue of the *Stachel*, the GPS environment (Figs. 5A; 3B) has been optimized by evolution not only to enable or disable GPS cleavage but also to enable exposition of this agonist[60], likely in response to mechanical stimuli for many aGPCR[61–69]. As noted since the characterization of the first GAIN domain structure[4], the apparently tight association of this sequence as a central β-strand in the GAIN subdomain B is puzzling, also considering the high thermal stability of GAIN domains. The *h*B2-HG construct has a melting point of $76.60 \pm 0.08\,°C$ (Supplementary Fig. 3), and similar values have been observed by us for other GAIN domains. The observed flexibility of the flanking loop regions observed in MD simulations in this and previous studies[18] may contribute to the dynamic process of *Stachel* release. AF2 predictions[27] of most aGPCR indicate no significant interaction between the GAIN domain and the 7TM domain that would result in a stable association of these two domains. Furthermore, the GAIN domain could so far not be visualized in many cryo-EM studies of GAIN-7TM constructs[19–25], likely due to the flexible linkage of the two domains. This situation favors a "one-and-done" signaling mechanism (dissociation model), in which *Stachel* binding to the 7TM domain is preceded by dissociation of the GAIN domain, NTF release[60] and permanent receptor activation, only terminated by internalization[70]. *Stachel*-mediated signaling requiring autoproteolysis has been demonstrated for G1[71] or L3[72,73]. A variation of *Stachel*-dependent signaling is that the *Stachel* sequence is prebound in the 7TM domain binding pocket and undergoes isomerization after mechanoactivation or ligand binding[63] as has been observed in glycoprotein hormone receptors[74]. However, there is growing evidence that aGPCRs can also signal via a *Stachel*-independent mechanism, which is characterized by a direct coupling of mechanical stimuli from the ECR to the 7TM domain[10,75,76]. For ADGRL3, it was recently observed that the GAIN domain exhibits a constrained movement relative to the 7TM domain[10]. Further assays suggested that reorientations of the GAIN domain are directly coupled to the 7TM domain. It is also important to note in this context that aGPCRs such as ADGRG2 have been shown to signal via alternative pathways that are dependent or independent of GPS cleavage[77]. For ADGRB1, it has been demonstrated that constitutive signaling activity is present even in a construct that lacks the ectodomain and the *Stachel*[76].

*Stachel*-independent modulation of receptor activity is obviously one plausible signaling mechanism for cleavage-incompetent receptors. However, the binding mode of the *Stachel* to the 7TM domain observed in cryo-EM structures should also be possible for uncleaved receptors. In the case of B2, constitutive signaling activity was observed for a CTF construct but not the full-length receptor[78], supporting a *Stachel*-mediated activation mechanism. Such a *Stachel*-dependent GPCR signaling in uncleaved aGPCRs may operate via a reversible signaling mode, in which the *Stachel* sequence exists in an equilibrium between 7TM- and GAIN-bound states. Release of the *Stachel* from the GAIN domain likely requires the unfolding of the four terminal β-strands of GAIN subdomain B, whereas the residual part of this domain might maintain a folded state, supported by the high thermal stability of this part of the GAIN fold. The structural flexibility of the flaps flanking the GPS might support this unfolding step[18]. It is interesting to note that the highly conserved disulfide bridges link all four terminal β-strands and might ensure fast refolding kinetics in addition to the stabilization of the GPS environment for autoproteolysis in cleavage-competent receptors. In the cryo-EM structures of the cleavage-deficient F1 receptor[19] and a D1 construct that included the GAIN domain[79], the *Stachel* sequence was observed to be bound to the orthosteric pocket of the 7TM domain in a similar binding mode as in GPS-cleaved receptors. Our understanding of the extent and mechanism of GPS cleavage, as well as its relationship to *Stachel* release from the GAIN domain or to direct conformational coupling between the GAIN and 7TM domains, needs to be extended by studies of these highly dynamic processes. In this context, the role of the diverse and mobile flaps flanking the GPS is a fascinating aspect of future research.

## Methods

### Preparation of the *h*B2-HG construct

**Cloning.** For the *h*B2-HG construct, DNA encoding the ECR of human ADGRB2/BAI2 (aa 1-921) was synthesized by Thermo Fisher GeneArt. The desired insert (residues 528-921) was subcloned into a pHLsec vector[80] for transient transfection and small-scale expression tests, and subsequently into a modified PiggyBac transposon vector[81] for stable transfection and large-scale expression. For all cloning steps, a classical approach using PCR and restriction endonucleases was used. The final expression construct included an additional N-terminal hemagglutinin antibody-tag (HA-tag) and a C-terminal enteropeptidase recognition site followed by a Twin-Strep-tag® (TST).

**Cell culture.** HEK293S GnTI⁻ cells (obtained from Dr. Yvonne Jones, University of Oxford[82]) were cultured in Dulbecco's Modified Eagle Medium (DMEM; ThermoFisher Scientific, cat# 21969035) supplemented with 10 % (v/v) fetal bovine serum (FBS; Gibco cat# 10437028, Life technologies), 1 % (v/v) GlutaMAX (cat# 35050038, Life technologies) and 1 % (v/v) MEM nonessential amino acids (cat# 11140035, Life technologies) and kept in a humidified incubator at 37 °C balanced with 5 % $CO_2$. DMEM containing only 2 % (v/v) FBS was used during transient protein expression, and serum-free DMEM was used to prepare transfection mixtures. Our DMEM formulation for storing cells in liquid nitrogen contained 20 % (v/v) FBS and 10 % (v/v) DMSO (cat# D8418, Sigma Aldrich). All reagents were warmed up to 37 °C prior to usage. Stably transfected HEK293S GnTI⁻ cells were retrieved from cryo-storage and gradually scaled up over three passages until they were seeded in six roller bottles (cat# 681670, Greiner Bio-One), where they were maintained for 10 days with an additional medium exchange after 5 days. The conditioned medium from all passaging steps and the roller bottles was collected and pooled for purification.

**Stable cell line.** The PiggyBac Single Promotor vector (cat# PB531A-1, System Biosciences) was modified in-house to replace the gene encoding the red fluorescent protein (RFP) with a gene for green fluorescent protein (GFP) expression. In addition, the multiple cloning site was modified for subcloning via the *Eco*RI and *Xho*I restriction sites

from the pHLsec vectors. Stable cell lines were generated by cotransfection of HEK293S GnTI⁻ cells with the pCMV-hyPBase vector[83] followed by selection via addition of 5 μg/mL puromycin (cat# ant-pr-1, InvivoGen) over 10 days. Cells were then allowed to recuperate for approximately seven days before they were transferred into cryostorage until further use.

**Protein purification.** 1 mL BioLock biotin blocking solution (cat# 2-0205-050, IBA Lifesciences) was added per 1.5 L of conditioned medium and incubated for 2 h, gently stirring at 4 °C. The medium was then centrifuged for 30 min at 18,600 g and the supernatant filtered through a 0.22 μm polyethersulfone membrane (cat# 514-0332 P, Avantor). Prior to affinity chromatography, the medium was concentrated approximately 15-fold, using a hollow fiber cartridge with a surface area of 4800 cm² and a nominal molecular weight cutoff (NMWC) of 10 kDa (GE Healthcare). All chromatography procedures were carried out at 4 °C using either an ÄKTA express or ÄKTA pure chromatography system (GE Healthcare). 1 mL StrepTrap™ HP (cat# 28907546, Cytiva) was used for affinity chromatography and a HiLoad™ 16/600 Superdex™ 200 pg (cat# GE28-9893-35, Sigma Aldrich) for size exclusion chromatography (SEC). Before injection into the SEC column, samples were concentrated to a volume of approximately 2 mL using Amicon® Ultra centrifugal filters with an NMWC of 10 kDa (cat# UFC801024, Merck Millipore). Antibodies and materials used for western blotting are listed in Supplementary Table 5.

### Autoproteolysis assays using full-length constructs

**Cloning.** For cellular expression analyses of full-length receptors, cDNAs encoding hB2 33-1219 were first subcloned from the pHLsec vector into the pcDNA3.1(+) vector before mutagenesis. Mutations on hB2^S876F, hB2^H910A, hB2^S912A, hB3^R866H and all rL1 mutations were introduced by PCR and sequence-ligation-independent cloning approach. hB3^R866H/L821X was synthesized by Genscript.

**Cellular expression analyses of receptors.** HEK293T cells (obtained from ATCC, cat. No. CRL-3216) were cultured in DMEM supplemented with 10% fetal bovine serum and 1% Pen/Strep (Gibco™, cat# 15070063), and maintained in a humidified incubator at 37 °C and 5% CO₂. Cells were seeded into a 24-well cell culture plate (Greiner, Cat# 662160) one day before transfection, at which point 70–80% confluence was observed. Cells were then transiently transfected with receptor constructs using the Lipofectamine™ 2000 transfection system (ThermoFisher, cat# 11668019). Briefly, 400 ng of plasmid in serum-free DMEM was mixed with 1.6 μl of Lipofectamine™ 2000 reagent in serum-free DMEM. After 20 min, the transfection mixture was added into a well of cells refreshed with full DMEM (10% fetal bovine serum and 1% Pen/Strep). The cells were incubated for one day before lysis with 150 μl triton X-100 buffer (50 mM Tris, pH 8.0, 150 mM NaCl, 1% Triton X-100). After removing cell debris, lysates were mixed with Laemmli buffer, followed by SDS-PAGE and WB analyses. Antibodies used for western blotting are listed in Supplementary Table 5.

**Deglycosylation of hB2 33-1219 receptors.** HEK293T cells were seeded into 6-well cell culture plates (Greiner, Cat# 657160) one day before transfection. Cells were transiently transfected using Lipofectamine™ 2000 transfection system (ThermoFisher, cat# 11668019 similarly as aforementioned, with 1600 ng of plasmid and 6.4 μl of Lipofectamine™ 2000 reagent. Cells were lysed with 500 μl of Triton X-100 buffer one day after transfection. For examining N-glycosylation of receptors, 17 μl of the lysates was treated with 500 U PNGase F (NEB, cat# P0704S), 500 U Endo H (NEB, cat# P0702S) or distilled water (1 μl each), supplemented with 2 μl of their respective 10x commercial buffer. The mixture was incubated at room

temperature for 1 h. For full deglycosylation of receptors, 17 μl of the lysates was treated with 1 μl of Deglycosylation mix II (NEB, cat# P6044S) or distilled water, supplemented with 2 μl of their respective 10x commercial buffer. The mixture was first incubated at room temperature for 30 min and then incubated at 37 °C overnight. The reactions were terminated by adding Laemmli buffer. Samples were then analyzed by SDS-PAGE and Western blots.

### Autoproteolysis assays using HormR-GAIN ectodomain constructs

**Molecular cloning and protein expression.** DNA fragments encoding residues 470–849 of rL1 and residues 498–868 of hB3 were obtained by gene synthesis (Thermo Fisher GeneArt) and provided in vector pMA (ThermoFisher Scientific). Point mutations were generated by *QuikChange mutagenesis*. The coding regions were subcloned into the pHLsec[80] vector by PCR amplification using the AgeI - KpnI restriction sites. A C-terminal Fc-tag followed by a hexahistidin sequence was added to increase the mass of the CTF generated by GPS autocleavage. The primer and plasmid sequences are available as Supplementary Data 1. Expression vectors (pTwist CMV-Fc) containing the genes for the flap exchange constructs of hB2 (residues 528-921) and rL1 (residues 498–868) were obtained from Twist Bioscience (USA). Refer to the supplementary data fasta file for the sequence of the flap exchange variants. HEK293T cells were grown to 80-90% confluency in a T75 cell culture flask (cat# 658175, Greiner Bio-One) and transfected with 20 μg expression plasmid using polyethyleneimine (PEI) in a 1.5-fold excess over the expression plasmid mass. Protein expression was carried out for a period of two days at 37 °C in a humid atmosphere containing 5% CO₂.

**Protein purification.** For the constructs with a C-terminal His-tag elution buffer (50 mM Tris-HCl, pH 8; 300 mM NaCl; 500 mM imidazole) was added to the harvested expression medium to give a final imidazole concentration of 30 mM. 100 μl of washed His-bead slurry (Cat. No. 30410, Qiagen) was added, and the samples were incubated for one hour at 4 °C while shaking. The beads were collected by centrifugation at 10,000 × g. The beads were washed four times with 1 ml of wash buffer (50 mM Tris-HCl, pH 8; 300 mM NaCl; 30 mM imidazole) and the proteins were eluted by adding 250 μl of elution buffer and incubating 2 min at room temperature. After centrifugation, the supernatant was collected. For the constructs with a Twin-Strep-tag 100 μl of washed Strep-Tactin XT 4Flow slurry (Cat. No. 2-5010-002, IBA) was added to the harvested cell culture medium, and the sample was incubated for one hour at 4 °C with shaking. After centrifugation at 10,000 g for 10 min the beads were washed four times with 1 ml of wash buffer W (Cat. No. 2-1003-100, IBA) and the protein was eluted by adding 250 μl of buffer BXT (Cat. No. 2-1042-025, IBA) and incubating for 2 min at room temperature. After centrifugation, the supernatant was collected.

**Quantification of GPS cleavage.** The supernatant was analyzed by SDS-PAGE, followed by semi-dry western blotting on methanol-activated PVDF membranes (Cat. No. 10600023, Cytiva Amersham). After skim milk blocking and incubation with anti-His-HRP antibodies (cat.# 11965085001, Roche) or Strep-Tactin conjugated HRP (cat.# 2-1502-001; IBA), the proteins were detected using a luminol-peroxidase mixture (cat.# 10308449, Cytiva Amersham). Chemiluminescence was measured using a Biostep Celvin® S 420 chemiluminescence imager, and intensities were scaled using the program ImageJ[84] to a maximum pixel saturation of 65535. The resulting band intensities were quantified using the GelAnalyzer software (version 23.1.1, available at www.gelanalyzer.com) using the valley-to-valley analysis method. The cleavage fraction was determined as the intensity of the NTF band divided by the sum of the intensities of the bands of the NTF and uncleaved protein.

## GPS autocleavage kinetics

For the determination of the GPS cleavage kinetics of the ectodomains, 200 µl of the purified protein was incubated for 14 days at 37 °C in the elution buffer (see *Protein purification* above). At each timepoint (0, 1, 2, 4, 7, 10 and 14 days, with 0 days being directly after protein purification), a sample of 25 µl was collected and denatured by adding 6.25 µl of SDS sample buffer (250 mM Tris-HCl, pH 6.8; 50% (v/v) glycerol; 4% (w/v) SDS; 25% (v/v) β-mercaptoethanol; 0.25 % (w/v) bromophenol blue) and boiling at 95 °C for 10 min. Samples were stored at −20 °C until SDS-PAGE analysis for the determination of the GPS cleavage fractions. 1–5 µl of the sample volume was applied to SDS-PAGE analysis, depending on protein concentration. The gels were analyzed as described above (*Quantification of GPS cleavage*). To determine the kinetics of the reactions, the cleaved fractions were plotted against the reaction time. The k value and half-life of the reaction were determined by a first-order reaction fit (model one-phase decay with the plateau fixed at 100 % cleavage product) using GraphPad Prism (version 10.4.1 for Windows, GraphPad Software, Boston, Massachusetts, USA).

## Dynamic light scattering (DLS) studies

DLS measurements were performed using a DynaPro® NanoStar® instrument. Samples were centrifuged for 20 min at 21,130 g and 4 °C, transferred into either a disposable plastic cuvette or a reusable quartz cuvette and allowed to equilibrate at 20 °C in the instrument for 6 min before the measurement was started. Each measurement was averaged across 100 individual runs with an acquisition time of 5 s. Data for estimation of particle size of the different *h*B2 constructs (Supplementary Table 2) were collected in a buffer consisting of 25 mM Tris, 150 mM NaCl, pH 8.0 (4 °C), using a viscosity of 1.019 (20 °C) and a refraction index of 1.333 (589 nm, 20 °C).

## Crystal structure analysis

Diffraction quality crystals were obtained via hanging drop vapor diffusion by mixing 1 µL protein solution (5 mg/mL *h*B2-HG) with 1 µL reservoir solution, containing 100 mM sodium cacodylate, pH 6.5, 200 mM MgCl$_2$ and 20 % (w/v) PEG 8000. Crystals grew as clusters of thin plates, which were separated by applying gentle pressure using a nylon loop. Crystals were flash frozen and stored in liquid nitrogen. Data was collected using synchrotron radiation at EMBL beamline P13 at the PETRA III storage ring (DESY, Hamburg, Germany), at λ = 1.77121 Å by an EIGER 16 M detector. Crystals were kept at 100 K in a constant jet of nitrogen gas. XDS (BUILT 20210323)[85] was used for indexing, integration and scaling of reflections. STARANISO 2.3.74[86] was used to apply an anisotropic cut-off of merged intensity data. Final scaling and merging were carried out by AIMLESS 0.7.7[87]. Molecular replacement using the model 4dlo[4] and refinement were carried out with the PHENIX[88] suite (version 1.18.2_3874) while Coot 0.9.8.92[89] was used for manual model building. Figures were generated using PyMOL (www.pymol.org). The refined model has been deposited in the Protein Data Bank as 8oek.

## Molecular dynamics simulations

**ADGRL1 MD simulation.** A model of the uncleaved L1 GAIN domain was constructed based on the crystal structure of the cleaved HormR-GAIN domain of rat L1 (PDB ID: 4dlq)[4]. The geometry of the peptide bond between the L$^{GPS,-1}$ and T$^{GPS,+1}$ residues of the GPS was manually modelled using the *Builder* utility in PyMOL[90]. The CHARMM-GUI was used to normalize bond lengths and generate minimization and equilibration inputs using the CHARMM36 forcefield for GROMACS version 2020.2[91–94]. Protonation states assigned to the model were verified by pKa analysis via Karlsberg2 + [95]. The N-acetylglucosamine residues at N531, N640, N741, N800, N805, and N826 of the crystal structure were included in the model. Water boxes using the TIP3P water model[96] were generated with the CHARMM-GUI using a distance

padding of 10 Å and charge-neutralized with 0.15 M NaCl. After minimization using the steepest-descent method for 5000 steps, a 125,000-step equilibration with 1 fs timestep was performed to yield the equilibrated model. To obtain a starting structure that is competent for GPS hydrolysis, the unprotonated nitrogen atoms of the H$^{GPS,-1}$ residues and the oxygen atom of the T$^{GPS,+1}$ hydroxyl nucleophile was pulled together by a biased MD simulation. After 5 ns of equilibration with a time step of 1 fs, a harmonic potential was applied on both atoms with a force constant of 1000 kJ/mol/nm² and a pull rate of −0.001 nm/ns until the run terminated due to a low-distance warning (HSD: 510 ps, HSE: 650 ps). With the resulting configurations, an equilibration cascade with decreasing harmonic potential holding the N$_{\delta/\varepsilon}$-O$_\gamma$ distance constant for 100 ns with a decreasing force constant of 1000, 500, 200 and 100 kJ/mol/nm², while simultaneously applying backbone, side chain and dihedral restraints of 400, 400, 40 and 4 kJ/mol/nm² or kJ/mol/deg, respectively, on the protein. After equilibration, triplicate unbiased MD simulations for 1500 ns using the CHARMM36 force field in GROMACS were performed.

**ADGRB2 MD simulation.** The crystal structure of the B2 HormR-GAIN domains determined in this work was used as a template for the preparation of the MD model. The loop of residues 605–610, which was absent in the crystallographic model, was modeled using SuperLooper2[97]. The other, large missing section (residues 770–810) was deemed too large to be modeled and was left as a gap within the model. Buried waters within the structure were added using dowser[98]. MD system setup was carried out using CHARMM-GUI[91], treating the termini of the introduced gap as amidated and acetylated and the protein C- and N-termini as neutral. After evaluating residue pKa values using Karlsberg2+ with only H910 showing ambiguous pKa, three independent systems were set up with H910 as HSD, HSE and HSP, respectively. Glycosylations were added according to the UniprotKB entry O60241 at residues N560, N645 and N867. Water boxes were generated with CHARMM-GUI using a distance padding of 10 Å and charge-neutralized with 0.15 M NaCl. MD was performed using the CHARMM36 force field in GROMACS 2020.2. After minimization using the steepest-descent method for 5000 steps, three independent 125,000-step equilibration runs with 1 fs timestep were performed per model before starting production runs with 2 fs timestep for 1500 ns each.

Analysis of the trajectories was carried out with GROMACS 2020.2 and MDtraj within a Python3 environment and by using VMD[99]. Time-resolved pKa analysis was carried out via Karlsberg2 + [95], calculating residue pKa for frames taken every 10 ns from each trajectory (151 frames per replica).

## Conservation analysis

For comparison of the different aGPCR amino acid sequences, all available vertebrate sequences (608) were extracted from NCBI and the respective orthologous sequences were separately aligned with the program Muscle embedded in the sequence analysis program Unipro UGENE v. 45.1[100]. Phylogenetic outliers (e.g., wrongly annotated sequences), sequences with missing sequence fragments, and mis-alignments were deleted from the alignments and not considered further. The conservation of the individual amino acid positions was determined with Unipro UGENE v. 45.1.

## Statistics

Protein bands in black and white western blot images were quantified using ImageJ[84]. The cleaved percentage was calculated as the fraction of the intensity of the NTF band over the sum of the intensities of the NTF and full-length bands. The Shapiro-Wilk normality test was used to determine whether the data showed normal distribution. Data presented in Fig. 6C, F showed a normal, Gaussian distribution and was analyzed using an unpaired, two-tailed t-test. Data presented in Fig. 7C did not show a normal, Gaussian distribution for all datasets and was

analyzed using an unpaired, two-tailed Mann-Whitney test. Significance was analyzed in GraphPad Prism (version 10.4.1 for Windows, GraphPad Software, Boston, Massachusetts, USA). Exact *P*-values are given in Figs. 6C, F and 7C.

## Reporting summary
Further information on research design is available in the Nature Portfolio Reporting Summary linked to this article.

## Data availability
The crystal structure of the ADGRB2 HormR-GAIN domain structure has been deposited in the protein data bank under PDB ID 8OEK [https://doi.org/10.2210/pdb8OEK/pdb]. The data of the MD simulations and the images of gels and western blots are deposited at Zenodo under https://doi.org/10.5281/zenodo.15490168. All other data are presented in Supplementary Information files. Should any raw data files be needed in another format, they are available from the corresponding author upon request.

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

## Acknowledgements

We thank the Deutsche Forschungsgemeinschaft (DFG, German Research Foundation) for financial support through SFB1423, project number 421152132, subprojects A06 (T.L. and N.S.), B06 (T.L.), C01 and Z04 (P.H.), A05, and C04 (T.S.). We acknowledge the EMBL beamlines of the DESY synchrotron in Hamburg for beamtime and support. We thank the MX Laboratory at the Helmholtz Zentrum Berlin (BESSY II) for beam time as well as for travel support. We gratefully acknowledge the scientific support and HPC resources provided by the Erlangen National High Performance Computing Center (NHR@FAU) of the Friedrich-Alexander-Universität Erlangen-Nürnberg (FAU). Supported by the Open Access Publishing Fund of Leipzig University.

## Author contributions

Y.C. and R.S. contributed equally to this work. F.P.: *h*B2-HG protein preparation and crystal structure analysis; F.S.: MD simulations and analysis; Y.C.: GPS cleavage assays of full-length proteins; R.S.: GPS cleavage assays and kinetics of ectodomain constructs; B.K.: generation of most of the ectodomain expression constructs; T.S.: conservation analysis; T.L., P.H., N.S.: Conceptualization of the studies; T.L., P.H., N.S.: supervision of the studies. F.P. wrote the first draft of the manuscript with contributions from F.S., Y.C. and R.S. All authors contributed to the interpretations and paper writing.

## Funding

## Competing interests

The authors declare no competing interests.
