## [Transparent Peer Review file · Nature Communications]

Structural basis of GAIN domain autoproteolysis and cleavage-resistance in the adhesion G-protein coupled receptors

Corresponding Author: Professor Norbert Sträter

Version 0:

Reviewer comments:

Reviewer #1

(Remarks to the Author)

i think the revised version have welly addressed my previous questions. in the revised version the authors removed the unrelated dimer-monomer data and focused on the mechanism GPS cleavage. Some new data added in the revised version and clearly supported the conclusions. i have only two tiny points:

1. Organize the figure S6 to main figure as the alternative conformations is not visible previously, and it support the key role of the F/Y/W that missed in B2, which forms a key pai-pai interaction with H910.
2. There are still some redundant descriptions, for example in discussion section.

Reviewer #3

(Remarks to the Author)

The authors have clearly worked hard to address every major comment, through additional biochemical experiments, and substantial improvement in clarity and structure. The study now offers a coherent narrative linking structure and cleavage competence in adhesion GPCRs.

The authors now reference a companion QM/MM mechanistic preprint. This manuscript requires moderate revisions to be published as a stand-alone research article in Nat Comm (see review).

Reviewer #4

(Remarks to the Author)

1. My major concern with this manuscript is that the authors make the assumption that the lack of auto proteolysis of BA I2 in mammalian cell culture extends to in vivo systems. Previously it was shown that BA I3 was uncleaved when expressed in mammalian cells (thus the BAI3 structure was uncleaved) but endogenous BA I3 in brain homogenates was shown to be partially cleaved. BA I3 (and also BAI1) was immunoprecipitated using a BA I specific antibody, then they were visualized on western blot using other B AI1 and BA I3 specific antibodies. The authors mention these results as well as another paper that suggested BA I2 is activated by auto proteolysis at the end of the discussion, however they suggest that the observed bands that correspond to the cleavage product can be an alternative splice isoform or non specific species. I do not find their explanation likely. If the authors want to claim that BA I2 is uncleaved, they need to show that in vivo it is really uncleaved. Otherwise one of the most important takeaway from the paper (which is BAI2 is not cleaved in spite of a canonical cleavage site) will remain unsupported.

2. Furthermore, I noticed that the authors agree to the possibility of cleavage of BAI2 in their response to reviewers. "These fragments have been described previously in the literature and are referenced in our manuscript. We agree that further investigation is needed to determine whether they arise from GPS autoproteolysis, proteolysis near the GPS, or alternative splicing."(response to Rev 2 point 5)

And I believe that they agree that they observe cleavage in their own experiments: "the isolated hB2 ectodomain displayed low but detectable autocleavage." (response to Rev 3 point 6)

However, I do not think they write this clearly in the manuscript text. Instead manuscript abstract says: " We determined the crystal structure of the Hormone Receptor (HormR) and GAIN domains of ADGRB2/BAI2 and found that this aGPCR is resistant to autoproteolysis, despite possessing a canonical HLS sequence motif at the GPS."

Thus I believe the manuscript text was not corrected to the satisfaction of previous reviews.

3. Another important point that I believe the authors are making a wrong assumption about is this: which residues at the cleavage site are absolutely essential and which are not absolutely essential. The authors assume that the histidine in the typical cleavage sequence H X T/S is essential. They keep referring to it as the canonical HXT/S motif. The reference they give for it does not say Histidine is essential. It actually says Histidine can be another residue and cleavage can still happen. The authors probably think histidine is essential because it is the general base and it needs to start the reaction by removing the hydrogen from the hydroxyl group of the threonine/serine. Histidine is not absolutely needed because a water molecule (a hydroxyl) can also act as a general base to do the identical reaction that a histidine would do. This statement is based on other similar enzymatic reactions that enzymologists have studied over 50 years. In summary, auto proteolysis within the gain domain is chemically similar to the chemical reaction that classical serine proteases do. In serine proteases, histidine is used as a general base to remove the hydroxyl hydrogen from the serine but a water molecule can also do the same job although it would be less efficient. Thus histidine is not absolutely essential. In both BAI 1 and BAI 3 there is an arginine residue in place of histidine and they both are uncleaved in mammalian cells but they both are partially cleaved in brain homogenates as shown in the mentioned reference. The residue that is absolutely needed in serine proteases is actually the serine which can only be replaced by a threonine or a cysteine which both have a -OH or -SH group that can do the same job (which is nucleophilic attack on the carbonyl carbon). There is no reason why the chemistry should be different between serine proteases and GAIN domain proteolysis. Thus, the authors assumption about histidine being critical does not seem right. There are many studies that mutate the Histidine to abolish the cleavage. These studies are not a proof that histidine is essential and another residue cannot do the job. Here is a reference discussing atypical serine proteases. I found just one. There should be more examples. <https://pmc.ncbi.nlm.nih.gov/articles/PMC2590910/> (Please do not trust ChatGPT about information on serine proteases. It is not giving the right answer.) Arginine is commonly found instead of His in aGPCRs. There is no reason why adhesion GPCRs that have an R X S/T motif should not be cleaved. GAIN domain does not need to be a fast and efficient enzyme unlike most other serine proteases. It can do the catalysis very slowly so it does not need to have the most efficient catalytic triad.

4. I completely agree with the authors that the entire chemical environment within the GAIN domain is important for cleavage and I commend them for the work that they have done to reveal the other components that effect cleavage within the gain domain. I would suggest that they go through the manuscript and correct the wording that says BA I2 is not cleaved and that this is surprising. I believe that all BAIs are cleaved in vivo but their cleavage is not efficient. In the case of BAI 2 it is not efficient even with a histidine. The authors can easily modify the paper to say that they have identified the determinants that switch BAIs from less efficient enzymes that cleave partially in vivo to more efficient enzymes that cleave also in mammalian cells. They can say they have found ways of making them cleave more efficiently. However, I do not think they can claim BAIs are uncleaved to start with. (Please note that this claim will be very misleading for the field). If the authors want to claim BAI2 is uncleaved, they need to extend their work into in vivo systems, test their claims and really prove that BAI2 is not cleaved in vivo. If they claim the cleavage band observed in brain homogenates is an alternatively spliced isoform, then they need to find an antibody that binds to the N-terminal part of BAI2 and show that no N-terminal fragment exists while the C terminal fragment is there.

5. Another important concern, in the Discussion, is how the authors interpret the many results in the literature to come up with their model for how aGPCRs function. Based on the below text:
"AF2 predictions²⁷ of most aGPCR indicate no significant interaction between the GAIN domain and the 7TM domain that would result in a stable association of these two domains. Furthermore, the GAIN domain could so far not be visualized in many cryo-EM studies of GAIN-7TM constructs^{19,21,20,22,24,23,25}, likely due to flexible linkage of the two domains. This situation favors a "one-and-done" signaling mechanism (dissociation model), in which Stachel binding to the 7TM domain is preceded by dissociation of the GAIN domain, NTF release and permanent receptor activation, only terminated by internalization. Stachel-mediated signaling requiring autoproteolysis has been demonstrated for G166 or L3 67,68."
Although authors removed Fig 10 as requested by a reviewer and changed the discussion, the authors still disregard important data in the field and highlight other data, which is not a proof, in favor of the one-and-done mechanism as proof for it. I urge them to keep a balanced view and cite the literature without bias rather than focusing on one mechanism which their group has discovered.

As for cryo EM structures cited (19,21,20,22,24,23,25), minimal flexibility will be enough to wipe out the Gain domain density. If previous studies focused on high resolution at the TM domain, they will for sure not observe the density from the Gain domain. But it does not mean it is not there. There is a recent study which has extensively studied Gain-TM orientation using single molecule FRET and clearly showed that there are three conformational states of the gain and TM which are close to each other and the GAIN and TM domains do not move "flexible" as assumed by the authors. The cryo-EM structure also confirms the sm FRET data and activating antibodies correlate GAIN-TM conformations to different signaling states. Single molecule FRET experiments is the gold standard to study the dynamics of domains in other receptor systems and they should be in aGPCRs as well. I expect the authors to consider this data in their model. Furthermore, a paper that does not seem to be cited (Salzman PNAS) also suggest an extracellular domain mediated mechanism and shows receptor activation is independent of autoproteolysis and also independent of the critical Phe residue in the Stachel. This paper also needs to be considered.

From the tone of the authors (in the following paragraphs of the discussion), it reads like they believe Stachel peptide is

something made of magic rather than amino acids. If Stachel peptide can come out and bind to the TM and wait there to be isomerized etc, why cannot other amino acids in other places of the protein, for example in the extracellular loops or on the extracellular part do the same thing? Why does it have to be the Stachel peptide doing the activation? This assumption does not have a basis. This is unclear to me. I believe the authors are basing this assumption on the observation that their mutations on the TA of uncleaved receptors affected function, but these mutations could affect other properties of the receptor as well, such as protein dynamics, I do not think they ensure that no other function of the protein is affected. Thus "TA does everything idea" remains as an assumption.

6. One more important point is the expression test. The authors made an effort to test whether the mutations have affected the expression of the protein, however just because a protein is expressed does not mean it is properly folded. Because proper folding of the GAIN domain is essential for cleavage, a mutation that affects proper folding will also affect protein cleavage. I suggest that they should test proper cell-surface localization of mutants or use other methods to ensure proper folding of their mutant proteins.

I believe that these studies are valuable and provide depth into aGPCR cleavage. The work is sound in my opinion. However, I would like to see the "assumptions and immature claims" which may lead the field into possibly wrong directions to be removed before the paper is published. It is better to make no claims than wrong claims.

Version 1:

Reviewer comments:

Reviewer #4

(Remarks to the Author)
no additional comments.

Reply to the reviewers comments

First of all we would like to thank the reviewers for their time and effort and the constructive comments!

We have reviewed the entire manuscript concerning the points raised by the reviewers.

In addition to the revised documents, we also uploaded a manuscript version in which the changes to the previous submission are marked as track changes.

REVIEWER COMMENTS

Reviewer #1 (Remarks to the Author):

I think the revised version have welly addressed my previous questions. in the revised version the authors removed the unrelated dimer-monomer data and focused on the mechanism GPS cleavage. Some new data added in the revised version and clearly supported the conclusions. i have only two tiny points:

1. Organize the figure S6 to main figure as the alternative conformations is not visible previously, and it support the key role of the F/Y/W that missed in B2, which forms a key pai-pai interaction with H910.

Figure S6 shows the alternative conformations of H910 in the B2 crystal structure together with the electron density at two contour levels ("normal" level of 1.3σ and low level at 0.6σ). The alternative conformations of H910 and the electron density (at low contour level) were already shown in Figure 2C in the revision 1 manuscript, but the lower occupancy rotamer was represented in transparency and therefore not well visible. In Figure 2C we now represent the alternative conformer without transparency and in the same color as in Figure S6 (which has the purpose of demonstrating the density at high and low contour level in comparison) to improve its visibility.

2. There are still some redundant descriptions, for example in discussion section.

We reviewed the manuscript to make it more concise and to remove redundancy. We removed some information from the discussion that has already been mentioned in the results and that was not present in the context of additional discussion.

Reviewer #3 (Remarks to the Author):

The authors have clearly worked hard to address every major comment, through additional biochemical experiments, and substantial improvement in clarity and structure. The study now offers a coherent narrative linking structure and cleavage competence in adhesion GPCRs.

The authors now reference a companion QM/MM mechanistic preprint. This manuscript requires moderate revisions to be published as a stand-alone research article in Nat Comm (see review).

Reviewer #4 (Remarks to the Author):

1. My major concern with this manuscript is that the authors make the assumption that the lack of auto proteolysis of BAI2 in mammalian cell culture extends to *in vivo* systems. Previously it was shown that BAI3 was uncleaved when expressed in mammalian cells (thus the BAI3 structure was uncleaved) but endogenous BAI3 in brain homogenates was shown to be partially cleaved. BAI3 (and also BAI1) was immunoprecipitated using a BAI specific antibody, then they were visualized on western blot using other BAI1 and BAI3 specific antibodies. The authors mention these results as well as another paper that suggested BAI2 is activated by auto proteolysis at the end of the discussion, however they suggest that the observed bands that correspond to the cleavage product can be an alternative splice isoform or non specific species. I do not find their explanation likely. If the authors want to claim that BAI2 is uncleaved, they need to show that *in vivo* it is really uncleaved. Otherwise one of the most important takeaway from the paper (which is BAI2 is not cleaved in spite of a canonical cleavage site) will remain unsupported.

We thank the reviewer for this remark and carefully revised the whole manuscript accordingly to avoid the impression that B2 or B3 are proven to be cleavage incompetent receptors *in vivo*. While our study contributes significant new information on the structural determinants of autocleavage activity of GAIN domains and we investigated autocleavage using ectodomains and full-length receptors after recombinant expression in HEK293 cell culture, we cannot exclude the possibility that the very low cleavage activity of BAI2 is sufficient for function native tissue or possibly even enhanced, for example by different conditions in the secretory pathway or by interactions of the ectodomains after insertion in the plasma membrane.

However, we think that further work is needed to exclude the possibility that the CTF-size fragments observed in mouse brain tissue by Arac et al. for BAI1 and BAI3 (PMID: 22333914) and by Okajima for BAI2 (PMID: 20367554) are due to the expression of C-terminal aGPCR fragments. For example, mouse BAI3 contains a highly active internal promoter that generates CTF-like fragments (see PMID: 31363148, Supplementary Table S3 or Uniprot id Q8BJ13). In the western blots reported by Shiu et al. (PMID: 37337931), CTF-like fragments are observed also in mouse brain tissue of a mouse line carrying a 7-bp deletion in exon 10 of BAI3, which encodes part of the GAIN domain and renders autoproteolysis impossible, resulting in a frameshift and premature stop codon.

Another not yet widely entertained possibility for the occurrence of NTF-/CTF-sized fragments of aGPCRs that are seemingly unable to conduct GAIN domain-mediated self-cleavage due to the lack of canonical residues flanking the GPS, is their processing by proteases such as MMPs at extracellular sites close to the cell membrane and thus also to the bona fide GPS. These non-GAIN domain mediated proteolytic events may serve to resolve adhesive aGPCR-ligand interactions to free the receptor for internalisation and signal termination. The size of the resulting proteolysis-dependent fragments only minimally differ from GPS-derived fragments, and may only be found *in vivo* in the presence of the cognate protease as opposed to *in vitro* analyses, where only the receptor is expressed. This possibility was demonstrated at least once for the rat ADGRL1/Lphn1 receptor, which is cleaved at its GPS and in addition at a second site 15 residues C-terminal to the GPS (19161337). This results in a 2 kDa difference in molecular weight of the CTF, which may be hard to resolve using gel-based analyses. Similar events occurring *in vivo* at different self-cleaved or non-self-cleaved aGPCRs need to be considered in interpreting the fragment identities too and warrant care with the assignment of fragment identities *in vivo*. While it should be possible to discriminate between CTF-size fragments

generated by autoproteolysis and those generated from alternative promoters via antibodies targeting the NTF, it is very difficult to characterize cleavage by proteases in native tissue.

In summary, we revised the text to avoid statements suggesting that ADGRB receptors are cleavage-resistant. Instead, we discuss arguments for and against autoproteolysis in these receptors.

2. Furthermore, I noticed that the authors agree to the possibility of cleavage of BAI2 in their response to reviewers.

“These fragments have been described previously in the literature and are referenced in our manuscript. We agree that further investigation is needed to determine whether they arise from GPS autoproteolysis, proteolysis near the GPS, or alternative splicing.”(response to Rev 2 point 5)

And I believe that they agree that they observe cleavage in their own experiments: “the isolated hB2 ectodomain displayed low but detectable autocleavage.” (response to Rev 3 point 6)

However, I do not think they write this clearly in the manuscript text. Instead manuscript abstract says: “ We determined the crystal structure of the Hormone Receptor (HormR) and GAIN domains of ADGRB2/BAI2 and found that this aGPCR is resistant to autoproteolysis, despite possessing a canonical HLS sequence motif at the GPS.”

Thus I believe the manuscript text was not corrected to the satisfaction of previous reviews.

This point is related to the previous one above and the reviewer is correct in all assumptions or comments. The cited sentence in the abstract was written with respect to the X-ray structure, in which the GPS appears uncleaved. To avoid misunderstanding, we rephrased this part of the abstract:

"We determined the crystal structure of the Hormone Receptor (HormR) and GAIN domains of ADGRB2/BAI2 and found that the protein was not cleaved at the GPS, despite possessing a canonical HLS sequence motif"

For BAI3 we rephrased in the abstract: "Disruption of this interaction reduced autoproteolytic activity in the ADGRL1 receptor and restored cleavage competence in ADGRB3, which is not cleaved upon expression in HEK293 cells." (instead of "cleavage-deficient")

For further changes concerning this point, please see the manuscript version with track changes.

3. Another important point that I believe the authors are making a wrong assumption about is this: which residues at the cleavage site are absolutely essential and which are not absolutely essential. The authors assume that the histidine in the typical cleavage sequence H X T/S is essential. They keep referring to it as the canonical HXT/S motif. The reference they give for it does not say Histidine is essential. It actually says Histidine can be another residue and cleavage can still happen. The authors probably think histidine is essential because it is the general base and it needs to start the reaction by removing the hydrogen from the hydroxyl group of the threonine/serine. Histidine is not absolutely needed because a water molecule (a hydroxyl) can also act as a general base to do the identical reaction that a histidine would do. This statement is based on other similar enzymatic reactions that enzymologists have studied over 50 years. In summary, auto proteolysis within the gain domain is chemically similar to the chemical reaction

that classical serine proteases do. In serine proteases, histidine is used as a general base to remove the hydroxyl hydrogen from the serine but a water molecule can also do the same job although it would be less efficient. Thus histidine is not absolutely essential. In both BAI 1 and BAI 3 there is an arginine residue in place of histidine and they both are uncleaved in mammalian cells but they both are partially cleaved in brain homogenates as shown in the mentioned reference. The residue that is absolutely needed in serine proteases is actually the serine which can only be replaced by a threonine or a cysteine which both have a -OH or -SH group that can do the same job (which is nucleophilic attack on the carbonyl carbon). There is no reason why the chemistry should be different between serine proteases and GAIN domain proteolysis. Thus, the authors assumption about histidine being critical does not seem right. There are many studies that mutate the Histidine to abolish the cleavage. These studies are not a proof that histidine is essential and another residue cannot do the job. Here is a reference discussing atypical serine proteases. I found just one. There should be more examples.

<https://pmc.ncbi.nlm.nih.gov/articles/PMC2590910/>

(Please do not trust ChatGPT about information on serine proteases. It is not giving the right answer.) Arginine is commonly found instead of His in aGPCRs. There is no reason why adhesion GPCRs that have an R X S/T motif should not be cleaved. GAIN domain does not need to be a fast and efficient enzyme unlike most other serine proteases. It can do the catalysis very slowly so it does not need to have the most efficient catalytic triad.

We agree with the statement that the histidine is not essential for the autoproteolysis reaction, and also with the comparison to serine proteases. In these enzymes even the serine nucleophile is not essential for activity as has been shown for several representatives (e.g. PUBMED ids 28754951, 10739913, 10712933). For a SEA domain, it was also shown that the serine nucleophile is not essential for *cis*-autoproteolytic activity. We checked the complete manuscript to avoid the impression that any residue of the GPS is essential for activity and instead refer to H-X/T-S as the canonical motif for efficient autoproteolytic cleavage.

Very low cleavage activity was observed for the human BAI2 ectodomain in our study, and a comparison of the GPS sequence to its orthologs suggests even much lower cleavage activity. In 608 BAI2 orthologs, His is found in only 31.6 %, Arg in 30.8 % and the remaining 38 % carry Lys, Ser, Tyr, or Pro. At the S/T position of the GPS, 18.6 % of the orthologs contain Ala. The loss of a suitable base or of the nucleophile likely results in even much lower cleavage activity. In 541 BAI1 orthologs, an arginine is present in 96%, but in the remaining 4 % this position is occupied by Lys, Asn, Gln, Ser, or Thr. Only in BAI3 are both the Arg and the S/T residues 100 % conserved across 653 orthologs. In none of the family members do we observe any indication of neutral drift (pseudogenization) that might explain such variability. We wonder, if cleavage is indeed physiologically essential, why are cleavage-efficient GPS motifs not conserved across orthologs, in particular for BAI2?

4. I completely agree with the authors that the entire chemical environment within the GAIN domain is important for cleavage and I commend them for the work that they have done to reveal the other components that effect cleavage within the gain domain. I would suggest that they go through the manuscript and correct the wording that says BAI2 is not cleaved and that this is surprising. I believe that all BAIs are cleaved in vivo but their cleavage is not efficient. In the case of BAI 2 it is not efficient even with a histidine. The authors can easily modify the paper to say that they have identified the determinants that switch BAIs from less efficient enzymes that cleave partially in vivo to more efficient enzymes that cleave also in mammalian cells. They

can say they have found ways of making them cleave more efficiently. However, I do not think they can claim BAIs are uncleaved to start with. (Please note that this claim will be very misleading for the field). If the authors want to claim BAI2 is uncleaved, they need to extend their work into in vivo systems, test their claims and really prove that BAI2 is not cleaved in vivo. If they claim the cleavage band observed in brain homogenates is an alternatively spliced isoform, then they need to find an antibody that binds to the N-terminal part of BAI2 and show that no N-terminal fragment exists while the C terminal fragment is there.

We have checked and changed the complete manuscript to avoid the impression that the BAI receptors are proven to be cleavage-incompetent. It is perfectly possible that the different conditions in brain tissue accelerate autoproteolysis of the BAI receptors and that the observed CTF-size fragments are indeed the result of autoproteolysis. As mentioned above it should also be considered that they may have derived from alternative promoters. For 67 % of all aGPCR genes, mRNA transcripts of CTFs without or with a short N-terminal extension have been observed, including BAI3. It is difficult to judge, which scenario is more likely. We do not claim that the fragments observed in brain tissue are derived from alternative promoters. This part of the discussion has been rephrased accordingly. Please refer to the redacted manuscript.

5. Another important concern, in the Discussion, is how the authors interpret the many results in the literature to come up with their model for how aGPCRs function. Based on the below text:

“AF2 predictions²⁷ of most aGPCR indicate no significant interaction between the GAIN domain and the 7TM domain that would result in a stable association of these two domains. Furthermore, the GAIN domain could so far not be visualized in many cryo-EM studies of GAIN-7TM constructs^{19,21,20,22,24,23,25}, likely due to flexible linkage of the two domains. This situation favors a "one-and-done" signaling mechanism (dissociation model), in which Stachel binding to the 7TM domain is preceded by dissociation of the GAIN domain, NTF release and permanent receptor activation, only terminated by internalization. Stachel-mediated signaling requiring autoproteolysis has been demonstrated for G166 or L3 67,68.”

Although authors removed Fig 10 as requested by a reviewer and changed the discussion, the authors still disregard important data in the field and highlight other data, which is not a proof, in favor of the one-and-done mechanism as proof for it. I urge them to keep a balanced view and cite the literature without bias rather than focusing on one mechanism which their group has discovered.

As for cryo EM structures cited (19,21,20,22,24,23,25), minimal flexibility will be enough to wipe out the Gain domain density. If previous studies focused on high resolution at the TM domain, they will for sure not observe the density from the Gain domain. But it does not mean it is not there. There is a recent study which has extensively studied Gain-TM orientation using single molecule FRET and clearly showed that there are three conformational states of the gain and TM which are close to each other and the GAIN and TM domains do not move “flexible” as assumed by the authors. The cryo-EM structure also confirms the sm FRET data and activating antibodies correlate GAIN-TM conformations to different signaling states. Single molecule FRET experiments is the gold standard to study the dynamics of domains in other receptor systems and they should be in aGPCRs as well. I expect the authors to consider this data in their model. Furthermore, a paper that does not seem to be cited (Salzman PNAS) also suggest an extracellular domain mediated mechanism and shows receptor activation is independent of autoproteolysis and also independent of the critical Phe residue in the Stachel. This paper also needs to be considered.

From the tone of the authors (in the following paragraphs of the discussion), it reads like they believe Stachel peptide is something made of magic rather than amino acids. If Stachel peptide can come out and bind to the TM and wait there to be isomerized etc, why cannot other amino acids in other places of the protein, for example in the extracellular loops or on the extracellular part do the same thing? Why does it have to be the Stachel peptide doing the activation? This assumption does not have a basis. This is unclear to me. I believe the authors are basing this assumption on the observation that their mutations on the TA of uncleaved receptors affected function, but these mutations could affect other properties of the receptor as well, such as protein dynamics, I do not think they ensure that no other function of the protein is affected. Thus "TA does everything idea" remains as an assumption.

We thank the reviewer for alerting us to the impression that our previous manuscript version does not reflect the current state of the art on aGPCR activation. In consequence, we have revised the discussion to present the *Stachel*-independent signaling mode in more detail and we added references that contributed evidence for these signaling modes, which beyond the experimental evidence are a plausible scenario how stimuli can be directly transmitted via the GAIN domain to the 7TM domain. The Kordon et al. 2024 study is most prominent in applying novel and powerful methods to study such mechanisms. In addition to this, we also referenced the Salzman et al. 2017 and the Kishore et al. 2016 paper, which support this signaling mode.

6. One more important point is the expression test. The authors made an effort to test whether the mutations have affected the expression of the protein, however just because a protein is expressed does not mean it is properly folded. Because proper folding of the GAIN domain is essential for cleavage, a mutation that affects proper folding will also affect protein cleavage. I suggest that they should test proper cell-surface localization of mutants or use other methods to ensure proper folding of their mutant proteins.

For the ectodomains, proper folding of the secreted protein is difficult to assess. For the kinetic experiments with these protein preparations, the time course of the cleavage reaction was fitted by a model, in which the plateau was fixed to 100 % cleavage. These curves did not show a clear indication that part of the secreted protein was autocleavage-inactive due to misfolding, which would be indicated by reaching a plateau of <100 % cleavage. However, for slow-cleaving constructs, we would not recognize deviation from a 100 % plateau level within the 14-day time interval. For the cleavage assays of the full-length receptors, the analyzed protein was derived from whole cell lysis and it may contain misfolded protein. To address the reviewer's point, we analyzed the glycosylation of the bands by digestions with different endoglycosidases to assess the fraction of immature N-glycosylated receptors, indicating potentially misfolded protein stuck in the ER (see Figures S13, S14 and S16). Lack of efficient cleavage of BAI2 in HEK293 cells was also observed previously by the Hall lab (PMID: 28891236).

I believe that these studies are valuable and provide depth into aGPCR cleavage. The work is sound in my opinion. However, I would like to see the "assumptions and immature claims" which may lead the field into possibly wrong directions to be removed before the paper is published. It is better to make no claims than wrong claims.

We revised the complete manuscript focusing on the points raised by the reviewer.

Additional changes required by journal policy

We shortened the abstract to less than 200 words.

The analysis of sequence conservation of the GPS sequence if B2, B3 and L1 was updated with a larger set of homologs. This resulted in different numbers but was inline with the previous results.